# Neural learning rules for generating flexible predictions and computing the successor representation

**Ching Fang[1], Dmitriy Aronov[1], LF Abbott[1], Emily L Mackevicius[1,2]***

[1]Zuckerman Institute, Department of Neuroscience, Columbia University, New York, United States; [2]Basis Research Institute, New York, United States

**Abstract** The predictive nature of the hippocampus is thought to be useful for memory-guided cognitive behaviors. Inspired by the reinforcement learning literature, this notion has been formalized as a predictive map called the successor representation (SR). The SR captures a number of observations about hippocampal activity. However, the algorithm does not provide a neural mechanism for how such representations arise. Here, we show the dynamics of a recurrent neural network naturally calculate the SR when the synaptic weights match the transition probability matrix. Interestingly, the predictive horizon can be flexibly modulated simply by changing the network gain. We derive simple, biologically plausible learning rules to learn the SR in a recurrent network. We test our model with realistic inputs and match hippocampal data recorded during random foraging. Taken together, our results suggest that the SR is more accessible in neural circuits than previously thought and can support a broad range of cognitive functions.

## Editor's evaluation

This important work provides compelling evidence for the biological plausibility of the Successor Representation (SR) algorithm. The SR is a leading computational hypothesis to explore whether neural representations are consistent with the hypothesis that the neural networks in specific brain areas perform predictive computations. Establishing a biologically plausible learning rule for SR representations to form is of high significance in the field of neuroscience.

*For correspondence:
em3406@columbia.edu

**Competing interest:** The authors declare that no competing interests exist.

## Introduction

To learn from the past, plan for the future, and form an understanding of our world, we require memories of personal experiences. These memories depend on the hippocampus for formation and recall (*Scoville and Milner, 1957*; *Penfield and Milner, 1958*; *Corkin, 2002*), but an algorithmic and mechanistic understanding of memory formation and retrieval in this region remains elusive. From a computational perspective, a key function of memory is to use past experiences to inform predictions of possible futures (*Bubic et al., 2010*; *Wayne et al., 2018*; *Whittington et al., 2020*; *Momennejad, 2020*). This suggests that hippocampal memory is stored in a way that is particularly suitable for forming predictions. Consistent with this hypothesis, experimental work has shown that, across species and tasks, hippocampal activity is predictive of the future experience of an animal (*Skaggs and McNaughton, 1996*; *Lisman and Redish, 2009*; *Mehta et al., 1997*; *Payne et al., 2021*; *Muller and Kubie, 1989*; *Pfeiffer and Foster, 2013*; *Schapiro et al., 2016*; *Garvert et al., 2017*). Furthermore, theoretical work has found that models endowed with predictive objectives tend to resemble hippocampal activity (*Blum and Abbott, 1996*; *Mehta et al., 2000*; *Stachenfeld et al., 2017*; *Momennejad et al., 2017*; *Geerts et al., 2020*; *Recanatesi et al., 2021*; *Whittington et al.,*

**eLife digest** Memories are an important part of how we think, understand the world around us, and plan out future actions. In the brain, memories are thought to be stored in a region called the hippocampus. When memories are formed, neurons store events that occur around the same time together. This might explain why often, in the brains of animals, the activity associated with retrieving memories is not just a snapshot of what happened at a specific moment-- it can also include information about what the animal might experience next. This can have a clear utility if animals use memories to predict what they might experience next and plan out future actions.

Mathematically, this notion of predictiveness can be summarized by an algorithm known as the successor representation. This algorithm describes what the activity of neurons in the hippocampus looks like when retrieving memories and making predictions based on them. However, even though the successor representation can computationally reproduce the activity seen in the hippocampus when it is making predictions, it is unclear what biological mechanisms underpin this computation in the brain.

Fang et al. approached this problem by trying to build a model that could generate the same activity patterns computed by the successor representation using only biological mechanisms known to exist in the hippocampus. First, they used computational methods to design a network of neurons that had the biological properties of neural networks in the hippocampus. They then used the network to simulate neural activity. The results show that the activity of the network they designed was able to exactly match the successor representation. Additionally, the data resulting from the simulated activity in the network fitted experimental observations of hippocampal activity in Tufted Titmice.

One advantage of the network designed by Fang et al. is that it can generate predictions in flexible ways,. That is, it canmake both short and long-term predictions from what an individual is experiencing at the moment. This flexibility means that the network can be used to simulate how the hippocampus learns in a variety of cognitive tasks. Additionally, the network is robust to different conditions. Given that the brain has to be able to store memories in many different situations, this is a promising indication that this network may be a reasonable model of how the brain learns.

The results of Fang et al. lay the groundwork for connecting biological mechanisms in the hippocampus at the cellular level to cognitive effects, an essential step to understanding the hippocampus, as well as its role in health and disease. For instance, their network may provide a concrete approach to studying how disruptions to the ways neurons make and break connections can impair memory formation. More generally, better models of the biological mechanisms involved in making computations in the hippocampus can help scientists better understand and test out theories about how memories are formed and stored in the brain.

---

*2020*; *George et al., 2021*). Thus, it is clear that predictive representations are an important aspect of hippocampal memory.

Inspired by work in the reinforcement learning (RL) field, these observations have been formalized by describing hippocampal activity as a predictive map under the successor representation (SR) algorithm (*Dayan, 1993*; *Gershman et al., 2012*; *Stachenfeld et al., 2017*). Under this framework, an animal's experience in the world is represented as a trajectory through some defined state space, and hippocampal activity predicts the future experience of an animal by integrating over the likely states that an animal will visit given its current state. This algorithm further explains how, in addition to episodic memory, the hippocampus may support relational reasoning and decision making (*Recanatesi et al., 2021*; *Mattar and Daw, 2018*), consistent with differences in hippocampal representations in different tasks (*Markus et al., 1995*; *Jeffery, 2021*). The SR framework captures many experimental observations of neural activity, leading to a proposed computational function for the hippocampus (*Stachenfeld et al., 2017*).

While the SR algorithm convincingly argues for a computational function of the hippocampus, it is unclear what biological mechanisms might compute the SR in a neural circuit. Thus, several relevant questions remain that are difficult to probe with the current algorithm. What kind of neural architecture should one expect in a region that can support this computation? Are there distinct forms of plasticity and neuromodulation needed in this system? What is the structure of hippocampal inputs

to be expected? A biologically plausible model can explore these questions and provide insight into both mechanism and function (*Marr and Poggio, 1976*; *Frank, 2015*; *Love, 2021*).

In other systems, it has been possible to derive biological mechanisms with the goal of achieving a particular network function or property (*Zeldenrust et al., 2021*; *Karimi et al., 2022*; *Pehlevan et al., 2017*; *Olshausen and Field, 1996*; *Burbank, 2015*; *Aitchison et al., 2021*; *Földiák, 1990*; *Tyulmankov et al., 2022*). Key to many of these models is the constraint that learning rules at any given neuron can only use information local to that neuron. A promising direction towards such a neural model of the SR is to use the dynamics of a recurrent neural network (RNN) to perform SR computations (*Vértes and Sahani, 2019*; *Russek et al., 2017*). An RNN model is particularly attractive as the hippocampus is highly recurrent, and its connectivity patterns are thought to support associative learning and recall (*Gardner-Medwin, 1976*; *McNaughton and Morris, 1987*; *Marr et al., 1991*; *Liu et al., 2012*). However, an RNN model of the SR has not been tied to neural learning rules that support its operation and allow for testing of specific hypotheses.

Here, we show that an RNN with local learning rules and an adaptive learning rate exactly calculates the SR at steady state. We test our model with realistic inputs and make comparisons to neural data. In addition, we compare our results to the standard SR algorithm with respect to the speed of learning and the learned representations in cases where multiple solutions exist. Our work provides

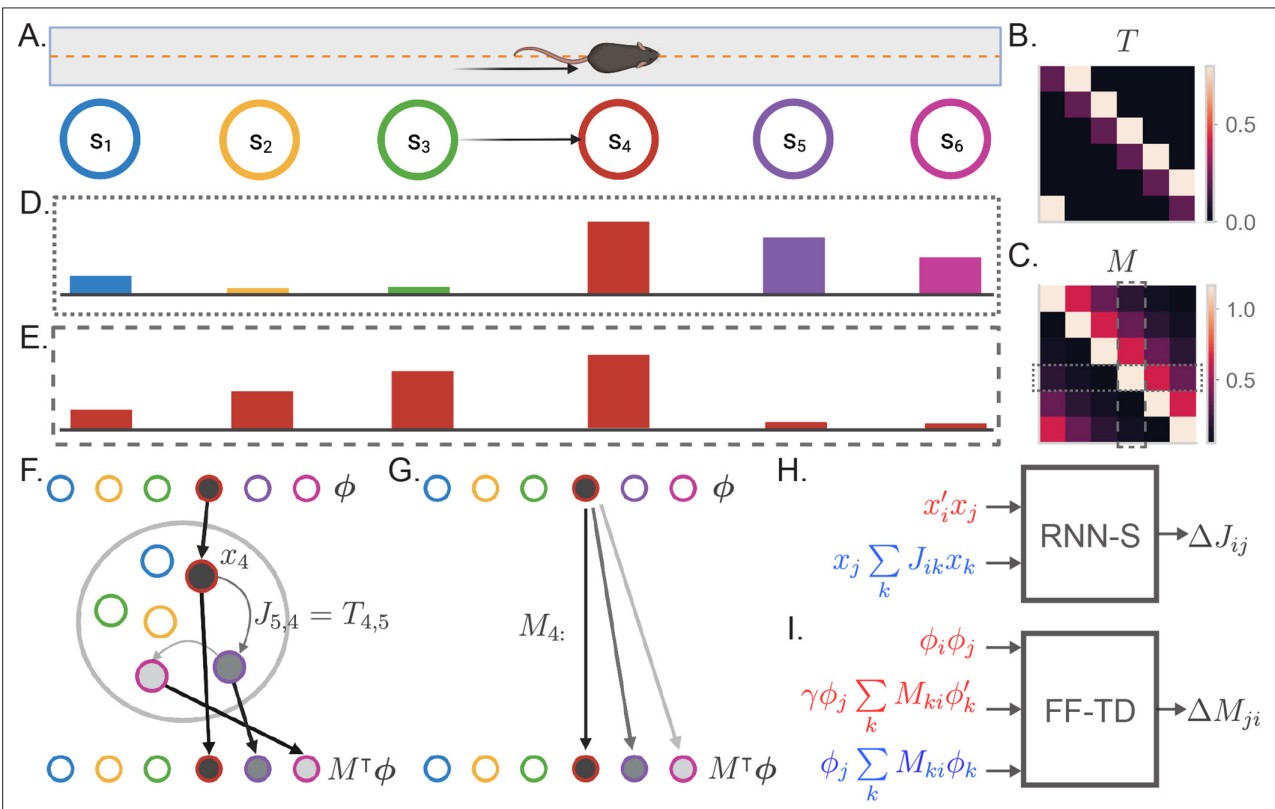

**Figure 1.** The successor representation and an analogous recurrent network model. (**A**) The behavior of an animal running down a linear track can be described as a transition between discrete states where the states encode spatial location. (**B**) By counting the transitions between different states, the behavior of an animal can be summarized in a transition probability matrix $T$. (**C**) The successor representation matrix is defined as $M = \sum_{i=0}^{\infty} \gamma^t T^t$. Here, $M$ is shown for $\gamma = 0.6$. Dashed boxes indicate the slices of $M$ shown in (**D**) and (**E**). (**D**) The fourth row of the $M$ matrix describes the activity of each state-encoding neuron when the animal is at the fourth state. (**E**) The fourth column of the $M$ matrix describes the place field of the neuron encoding the fourth state. (**F**) Recurrent network model of the SR (RNN-S). The current state of the animal is one-hot encoded by a layer of input neurons. Inputs connect one-to-one onto RNN neurons with synaptic connectivity matrix $J = T^\mathsf{T}$. The activity of the RNN neurons are represented by $x$. SR activity is read out from one-to-one connections from the RNN neurons to the output neurons. The example here shows inputs and outputs when the animal is at state 4. (**G**) Feedforward neural network model of the SR (FF-TD). The $M$ matrix is encoded in the weights from the input neurons to the output layer neurons, where the SR activity is read out. (**H**) Diagram of the terms used for the RNN-S learning rule. Terms in red are used for potentiation while terms in blue are used for normalization (*Equation 4*). (**I**) As in (**H**) but for the feedforward-TD model (*Equation 11*). To reduce the notation indicating time steps, we use $\prime$ in place of $(t)$ and no added notation for $(t - 1)$.

a mechanistic account for an algorithm that has been frequently connected to the hippocampus, but could only be interpreted at an algorithmic level. This network-level perspective allows us to make specific predictions about hippocampal mechanisms and activity.

## Results

### The successor representation

The SR algorithm described in *Stachenfeld et al., 2017* first discretizes the environment explored by an animal (whether a physical or abstract space) into a set of $n$ states that the animal transitions through over time (*Figure 1A*). The animal's behavior can then be thought of as a Markov chain with a corresponding transition probability matrix $T_{N \times N}$ (*Figure 1B*). $T$ gives the probability that the animal transitions to a state $s'$ from the state $s$ in one time step: $T_{ji} = P(s' = i | s = j)$. The SR matrix is defined as

$$M = \sum_{t=0}^{\infty} \gamma^t T^t = (I - \gamma T)^{-1} \tag{1}$$

Here, $\gamma \in (0, 1)$ is a temporal discount factor. $M_{ji}$ can be seen as a measure of the occcupancy of state $i$ over time if the animal starts at state $j$, with $\gamma$ controlling how much to discount time steps in the future (*Figure 1C*). The SR of state $j$ is the $j$th row of $M$ and represents the states that an animal is likely to transition to from state $j$. *Stachenfeld et al., 2017* demonstrate that, if one assumes each state drives a single neuron, the SR of $j$ resembles the population activity of hippocampal neurons when the animal is at state $j$ (*Figure 1D*). They also show that the $i$th column of $M$ resembles the place field (activity as a function of state) of a hippocampal neuron representing state $i$ (*Figure 1E*). In addition, the $i$th column of $M$ shows which states are likely to lead to state $i$.

### Recurrent neural network computes SR at steady state

We begin by drawing connections between the SR algorithm (*Stachenfeld et al., 2017*) and an analogous neural network architecture. The input to the network encodes the current state of the animal and is represented by a layer of input neurons (*Figure 1FG*). These neurons feed into the rest of the network that computes the SR (*Figure 1FG*). The SR is then read out by a layer of output neurons so that downstream systems receive a prediction of the upcoming states (*Figure 1FG*). We will first model the inputs $\phi$ as one-hot encodings of the current state of the animal (*Figure 1FG*). That is, each input neuron represents a unique state, and input neurons are one-to-one connected to the hidden neurons.

We first consider an architecture in which a recurrent neural network (RNN) is used to compute the SR (*Figure 1F*). Let us assume that the $T$ matrix is encoded in the synaptic weights of the RNN. In this case, the steady state activity of the network in response to input $\phi$ retrieves a row of the SR matrix, $M^T \phi$ (*Figure 1F*, subsection 4.14). Intuitively, this is because each recurrent iteration of the RNN progresses the prediction by one transition. In other words, the $t$th recurrent iteration raises $T$ to the $t$th power as in *Equation 1*. To formally derive this result, we first start by defining the dynamics of our RNN with classical rate network equations (*Amarimber, 1972*). At time $t$, the firing rate $\boldsymbol{x}(t)$ of $N$ neurons given each neurons' input $\phi(t)$ follows the discrete-time dynamics (assuming a step size $\Delta t = 1$)

$$\Delta \boldsymbol{x} = -\boldsymbol{x}(t) + \gamma J f(\boldsymbol{x}(t)) + \phi(t) \tag{2}$$

Here, $\gamma$ scales the recurrent activity and is a constant factor for all neurons. The synaptic weight matrix $J \in \mathcal{R}_{N \times N}$ is defined such that $J_{ij}$ is the synaptic weight from neuron $j$ to neuron $i$. Notably, this notation is transposed from what is used in RL literature, where conventions have the first index as the starting state. Generally, $f$ is some nonlinear function in *Equation 2*. For now, we will consider $f$ to be the identity function, rendering this equation linear. Under this assumption, we can solve for the steady state activity $\boldsymbol{x}_{ss}$ as

$$\boldsymbol{x}_{ss} = (I - \gamma J)^{-1} \phi \tag{3}$$

Equivalence between *Equation 1* and *Equation 3* is clearly reached when $J = T^{\mathsf{T}}$ (*Russek et al., 2017*; *Vértes and Sahani, 2019*). Thus, if the network can learn $T$ in its synaptic weight matrix, it will exactly compute the SR.

Here, the factor $\gamma$ represents the gain of the neurons in the network, which is factored out of the synaptic strengths characterized by $J$. Thus, $\gamma$ is an independently adjustable factor that can flexibly control the strength of the recurrent dynamics (see *Sompolinsky et al., 1988*). A benefit of this flexibility is that the system can retrieve successor representations of varying predictive strengths by modulating the gain factor $\gamma$. In this way, the predictive horizon can be dynamically controlled without any additional learning required. We will refer to the $\gamma$ used during learning of the SR as the baseline $\gamma$, or $\gamma_B$.

We next consider what is needed in a learning rule such that $J$ approximates $T^{\mathsf{T}}$. In order to learn a transition probability matrix, a learning rule must associate states that occur sequentially and normalize the synaptic weights into a valid probability distribution. We derive a learning rule that addresses both requirements (*Figure 1H*, Appendix 2),

$$\Delta J_{ij} = \eta x_i(t) x_j(t-1) - \eta x_j(t-1) \sum_k J_{ik} x_k(t-1),$$

(4)

where $\eta$ is the learning rate. The first term in *Equation 4* is a temporally asymmetric potentiation term that counts states that occur in sequence. This is similar to spike-timing dependent plasticity, or STDP (*Bi and Poo, 1998*; *Skaggs and McNaughton, 1996*; *Abbott and Blum, 1996*).

The second term in *Equation 4* is a form of synaptic depotentiation. Depotentiation has been hypothesized to be broadly useful for stabilizing patterns and sequence learning (*Földiák, 1990*; *Fiete et al., 2010*), and similar inhibitory effects are known to be elements of hippocampal learning (*Kullmann and Lamsa, 2007*; *Lamsa et al., 2007*). In our model, the depotentiation term in *Equation 4* imposes local anti-Hebbian learning at each neuron– that is, each column of $J$ is normalized independently. This normalizes the observed transitions from each state by the number of visits to that state, such that transition statistics are correctly captured. We note, however, that other ways of column-normalizing the synaptic weight matrix may give similar representations (Appendix 7).

Crucially, the update rule (*Equation 4*) uses information local to each neuron (*Figure 1H*), an important aspect of biologically plausible learning rules. We show that, in the asymptotic limit, the update rule extracts information about the inputs $\phi$ and learns $T$ exactly despite having access only to neural activity $x$ (Appendix 3). We will refer to an RNN using *Equation 4* as the RNN-Successor, or RNN-S. Combined with recurrent dynamics (*Equation 3*), RNN-S computes the SR exactly (*Figure 1H*).

As an alternative to the RNN-S model, we consider the conditions necessary for a feedforward neural network to compute the SR. Under this architecture, the $M$ matrix must be encoded in the weights from the input neurons to the hidden layer neurons (*Figure 1G*). This can be achieved by updating the synaptic weights with a temporal difference (TD) learning rule, the standard update used to learn the SR in the usual algorithm. Although the TD update learns the SR, it requires information about multiple input layer neurons to make updates for the synapse from input neuron $j$ to output neuron $i$ (*Figure 1I*). Thus, it is useful to explore other possible mechanisms that are simpler to compute locally. We refer to the model described in *Figure 1I* as the feedforward-TD (FF-TD) model. The FF-TD model implements the canonical SR algorithm.

## Evaluating SR learning by biologically plausible learning rules

To evaluate the effectiveness of the RNN-S learning rule, we tested its accuracy in learning the SR matrix for random walks. Specifically, we simulated random walks with different transition biases in a 1D circular track environment (*Figure 2A*). The RNN-S can learn the SR for these random walks (*Figure 2B*).

Because equivalence is only reached in the asymptotic limit of learning (i.e. $\Delta J \to 0$), our RNN-S model learns the SR slowly. In contrast, animals are thought to be able to learn the structure of an environment quickly (*Zhang et al., 2021*), and neural representations in an environment can also develop quickly (*Monaco et al., 2014*; *Sheffield and Dombeck, 2015*; *Bittner et al., 2015*). To remedy this, we introduce a dynamic learning rate that allows for faster normalization of the synaptic weight matrix, similar to the formula for calculating a moving average (Appendix 4). For each neuron, suppose that a trace $n$ of its recent activity is maintained with some time constant $\lambda \in (0, 1)$,

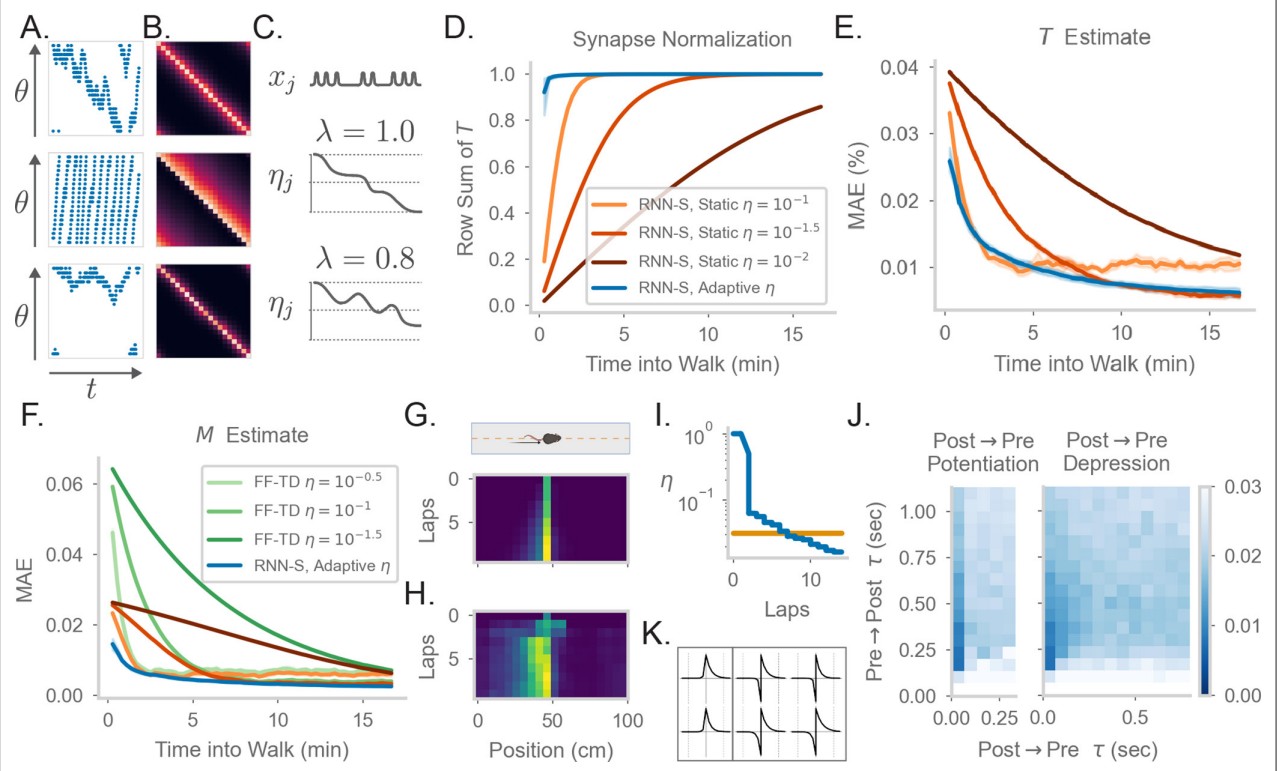

**Figure 2.** Comparing the effects of an adaptive learning rate and plasticity kernels in RNN-S. (A) Sample one-minute segments from random walks on a 1 meter circular track. Possible actions in this 1D walk are to move forward, stay in one place, or move backward. Action probabilities are uniform (top), biased to move forward (middle), or biased to stay in one place (bottom). (B) $M$ matrices estimated by the RNN-S model in the full random walks from (A).(C) The proposed learning rate normalization. The learning rate $\eta_j$ for synapses out of neuron $j$ changes as a function of its activity $x_j$ and recency bias $\lambda$. Dotted lines are at $[0.0, 0.5, 1.0]$. (D) The mean row sum of $T$ over time computed by the RNN-S with an adaptive learning rate (blue) or the RNN-S with static learning rates (orange). Darker lines indicate larger static learning rates. Lines show the average over 5 simulations from walks with a forward bias, and shading shows 95% confidence interval. A correctly normalized $T$ matrix should have a row sum of 1.0. (E) As in (D), but for the mean absolute error in estimating $T$. (F) As in (E), but for mean absolute error in estimating the real $M$, and with performance of FF-TD included, with darker lines indicating slower learning rates for FF-TD. (G) Lap-based activity map of a neuron from RNN-S with static learning rate $\eta = 10^{-1.5}$. The neuron encodes the state at 45cm on a circular track. The simulated agent is moving according to forward-biased transition statistics. (H) As in (G), but for RNN-S with adaptive learning rate. (I) The learning rate over time for the neuron in (G) (orange) and the neuron in (H) (blue). (J) Mean-squared error (MSE) at the end of meta-learning for different plasticity kernels. The pre→post (K+) and post→pre (K-) sides of each kernel were modeled by $Ae^{-\frac{1}{\tau}}$. Heatmap indices indicate the values $\tau$ s were fixed to. Here, K+ is always a positive function (i.e., $A$ was positive), because performance was uniformly poor when K+ was negative. K- could be either positive (left, "Post → Pre Potentiation") or negative (right, "Post → Pre Depression"). Regions where the learned value for $A$ was negligibly small were set to high errors. Errors are max-clipped at 0.03 for visualization purposes. 40 initializations were used for each K+ and K- pairing, and the heatmap shows the minimum error achieved over all intializations. (K) Plasticity kernels chosen from the areas of lowest error in the grid search from (J). Left is post → pre potentiation. Right is post → pre depression. Kernels are normalized by the maximum, and dotted lines are at one second intervals.

The online version of this article includes the following figure supplement(s) for figure 2:

**Figure supplement 1.** Comparing model performance in different random walks.

$$n(t) = \sum_{t' < t} \lambda^{(t-t')} x(t'), \tag{5}$$

If the learning rate of the outgoing synapses from each neuron $j$ is inversely proportional to $n_j$ $\left(\eta = \frac{1}{n_j(t)}\right)$, the update equation quickly normalizes the synapses to maintain a valid transition probability matrix (Appendix 4). Modulating synaptic learning rates as a function of neural activity is consistent with experimental observations of metaplasticity (*Abraham and Bear, 1996*; *Abraham, 2008*; *Hulme et al., 2014*). We refer to this as an adaptive learning rate and contrast it with the previous static learning rate. We consider the setting where $\lambda = 1$, so the learning rate monotonically decreases over time (*Figure 2C*). In general, however, the learning rate could increase or decrease over time if $\lambda < 1$

(*Figure 2C*), and $n$ could be reset, allowing for rapid learning. Our learning rule with the adaptive learning rate is the same as in *Equation 4*, with the exception that $\eta = \min(\frac{1}{n_j(t)}, 1.0)$ for synapses $J_{*j}$. This learning rule still relies only on information local to the neuron as in *Figure 1H*.

The RNN-S with an adaptive learning rate normalizes the synapses more quickly than a network with a static learning rate (*Figure 2D*, *Figure 2—figure supplement 1*) and learns $T$ faster (*Figure 2E*, *Figure 2—figure supplement 1*). The RNN-S with a static learning rate exhibits more of a tradeoff between normalizing synapses quickly (*Figure 2D*, *Figure 2—figure supplement 1A*) and learning $M$ accurately (*Figure 2E*, *Figure 2—figure supplement 1*). However, both versions of the RNN-S estimate $M$ more quickly than the FF-TD model (*Figure 2F*, *Figure 2—figure supplement 1*).

Place fields can form quickly, but over time the place fields may skew if transition statistics are consistently biased (*Stachenfeld et al., 2017*; *Monaco et al., 2014*; *Sheffield and Dombeck, 2015*; *Bittner et al., 2015*). The adaptive learning rate recapitulates both of these effects, which are thought to be caused by slow and fast learning processes, respectively. A low learning rate can capture the biasing of place fields, which develops over many repeated experiences. This is seen in the RNN-S with a static learning rate (*Figure 2G*). However, a high learning rate is needed for hippocampal place cells to develop sizeable place fields in one-shot. Both these effects of slow and fast learning can be seen in the neural activity of an example RNN-S neuron with an adaptive learning rate (*Figure 2H*). After the first lap, a sizeable field is induced in a one-shot manner, centered at the cell's preferred location. In subsequent laps, the place field slowly distorts to reflect the bias of the transition statistics (*Figure 2H*). The model is able to capture these learning effects because the adaptive learning rate transitions between high and low learning rates, unlike the static version (*Figure 2I*).

Thus far, we have assumed that the RNN-S learning rule uses pre→post activity over two neighboring time steps (*Equation 4*). A more realistic framing is that a convolution with a plasticity kernel determines the weight change at any synapse. We tested how this affects our model and what range of plasticity kernels best supports the estimation of the SR. We do this by replacing the pre→post potentiation term in *Equation 4* with a convolution:

$$\Delta J_{ij} = x_i(t) \sum_{t'=-\infty}^{t} K_+(t-t')x_j(t') + x_j(t) \sum_{t'=-\infty}^{t} K_-(t-t')x_i(t') - \eta x_j(t-1) \sum_k J_{ik}x_k(t-1) \tag{6}$$

In the above equation, the full kernel $K$ is split into a pre→post kernel ($K_+$) and a post→pre kernel ($K_-$). $K_+$ and $K_-$ are parameterized as independent exponential functions, $Ae^{-t/\tau}$.

To systematically explore the space of plasticity kernels that can be used to learn the SR, we performed a grid search over the sign and the time constants of the pre→post and post→pre sides of the plasticity kernels. For each fixed sign and time constant, we used an evolutionary algorithm to learn the remaining parameters that determine the plasticity kernel. We find that plasticity kernels that are STDP-like are more effective than others, although plasticity kernels with slight post→pre potentiation work as well (*Figure 2J*). The network is sensitive to the time constant and tends to find solutions for time constants around a few hundred milliseconds (*Figure 2JK*). Our robustness analysis indicates the timescale of a plasticity rule in such a circuit may be longer than expected by standard STDP, but within the timescale of changes in behavioral states. We note that this also contrasts with behavioral timescale plasticity (*Bittner et al., 2015*), which integrates over a window that is several seconds long. Finally, we see that even plasticity kernels with slightly different time constants may give results with minimal error from the SR matrix, even if they do not estimate the SR exactly (*Figure 2J*). This suggests that, although other plasticity rules could be used to model long-horizon predictions, the SR is a reasonable –although not strictly unique– model to describe this class of predictive representations.

### RNN-S can compute the SR with arbitrary $\gamma_R$ under a stable regime of $\gamma_B$

We next investigate how robust the RNN-S model is to the value of $\gamma$. Typically, for purposes of fitting neural data or for RL simulations, $\gamma$ will take on values as high as 0.9 (*Stachenfeld et al., 2017*; *Barreto et al., 2017*). However, previous work that used RNN models reported that recurrent dynamics become unstable if the gain $\gamma$ exceeds a critical value (*Sompolinsky et al., 1988*; *Zhang et al., 2021*). This could be problematic as we show analytically that the RNN-S update rule is effective only when the network dynamics are stable and do not have non-normal amplification (Appendix 2). If these conditions are not satisfied during learning, the update rule no longer optimizes for fitting the SR and the learned weight matrix will be incorrect.

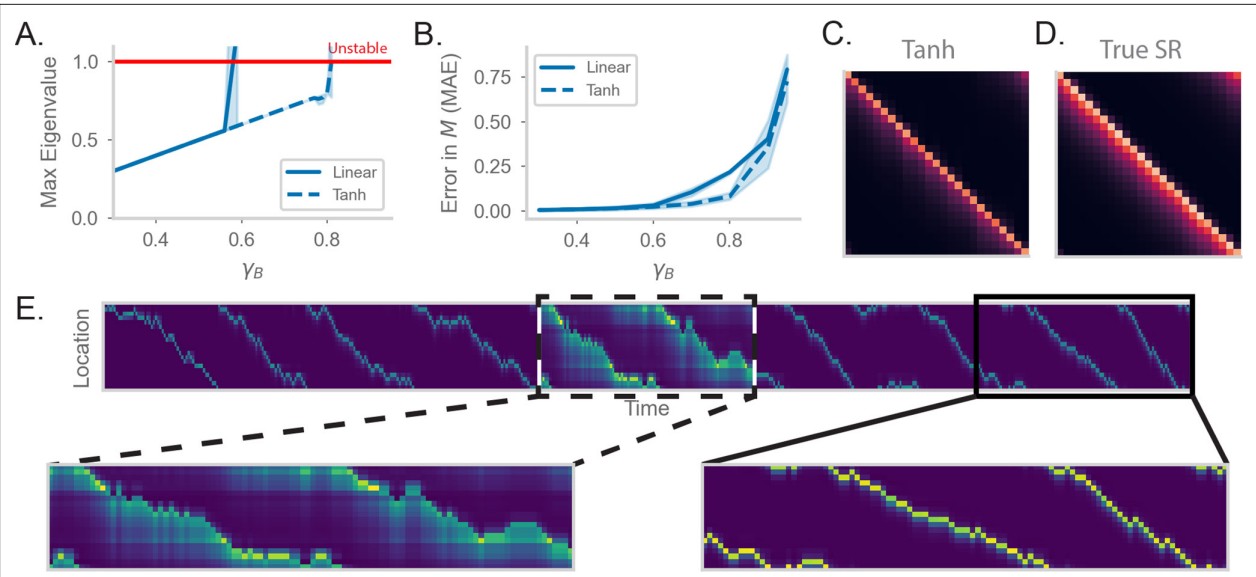

**Figure 3.** RNN-S requires a stable choice of $\gamma_B$ during learning, and can compute SR with any $\gamma_R$ (**A**) Maximum real eigenvalue of the $J$ matrix at the end of random walks under different $\gamma_B$. The network dynamics were either fully linear (solid) or had a tanh nonlinearity (dashed). Red line indicates the transition into an unstable regime. 45 simulations were run for each $\gamma_B$, line indicates mean, and shading shows 95% confidence interval. (**B**) MAE of $M$ matrices learned by RNN-S with different $\gamma_B$. RNN-S was simulated with linear dynamics (solid line) or with a tanh nonlinearity added to the recurrent dynamics (dashed line). Test datasets used various biases in action probability selection. (**C**) $M$ matrix learned by RNN-S with tanh nonlinearity added in the recurrent dynamics. A forward-biased walk on a circular track was simulated, and $\gamma_B = 0.8$. (**D**) The true $M$ matrix of the walk used to generate (**C**). (**E**) Simulated population activity over the first ten laps in a circular track with $\gamma_B = 0.4$. Dashed box indicates the retrieval phase, where learning is turned off and $\gamma_R = 0.9$. Boxes are zoomed in on three minute windows.

The online version of this article includes the following figure supplement(s) for figure 3:

**Figure supplement 1.** Understanding the effects of recurrency on stability.

We first test how the value of $\gamma_B$, the gain of the network during learning, affects the RNN-S dynamics. The dynamics become unstable when $\gamma_B$ exceeds 0.6 (*Figure 3—figure supplement 1*). Specifically, the eigenvalues of the synaptic weight matrix exceed the critical threshold for learning when $\gamma_B > 0.6$ (*Figure 3A*, 'Linear'). As expected from our analytical results, the stability of the network is tied to the network's ability to estimate $M$. RNN-S cannot estimate $M$ well when $\gamma_B > 0.6$ (*Figure 3B*, 'Linear'). We explored two strategies to enable RNN-S to learn at high $\gamma$.

One way to tame this instability is to add a saturating nonlinearity into the dynamics of the network. This is a feature of biological neurons that is often incorporated in models to prevent unbounded activity (*Dayan and Abbott, 2001*) Specifically, instead of assuming the network dynamics are linear ($f$ is the identity function in *Equation 2*), we add a hyperbolic tangent into the dynamics equation. This extends the stable regime of the network– the eigenvalues do not exceed the critical threshold until $\gamma_B > 0.8$ (*Figure 3A*). Similar to the linear case, the network with nonlinear dynamics fits $M$ well until the critical threshold for stability (*Figure 3b*). These differences are clear visually as well. While the linear network does not estimate $M$ well for $\gamma_B = 0.8$ (*Figure 3B*), the estimate of the nonlinear network (*Figure 3c*) is a closer match to the true $M$ (*Figure 3D*). However, there is a tradeoff between the stabilizing effect of the nonlinearity and the potential loss of accuracy in calculating $M$ with a nonlinearity (*Figure 3—figure supplement 1*).

We explore an alternative strategy for computing $M$ with arbitrarily high $\gamma$ in the range $0 \leq \gamma < 1$. We have thus far pushed the limits of the model in learning the SR for different $\gamma_B$. However, an advantage of our recurrent architecture is that $\gamma$ is a global gain modulated independently of the synaptic weights. Thus, an alternative strategy for computing $M$ with high $\gamma$ is to consider two distinct modes that the network can operate under. First, there is a learning phase in which the plasticity mechanism actively learns the structure of the environment and the model is in a stable regime (i.e. $\gamma_B$ is small). Separately, there is a retrieval phase during which the gain $\gamma_R$ of the network can be flexibly modulated. By changing the gain, the network can compute the SR with arbitrary prediction horizons,

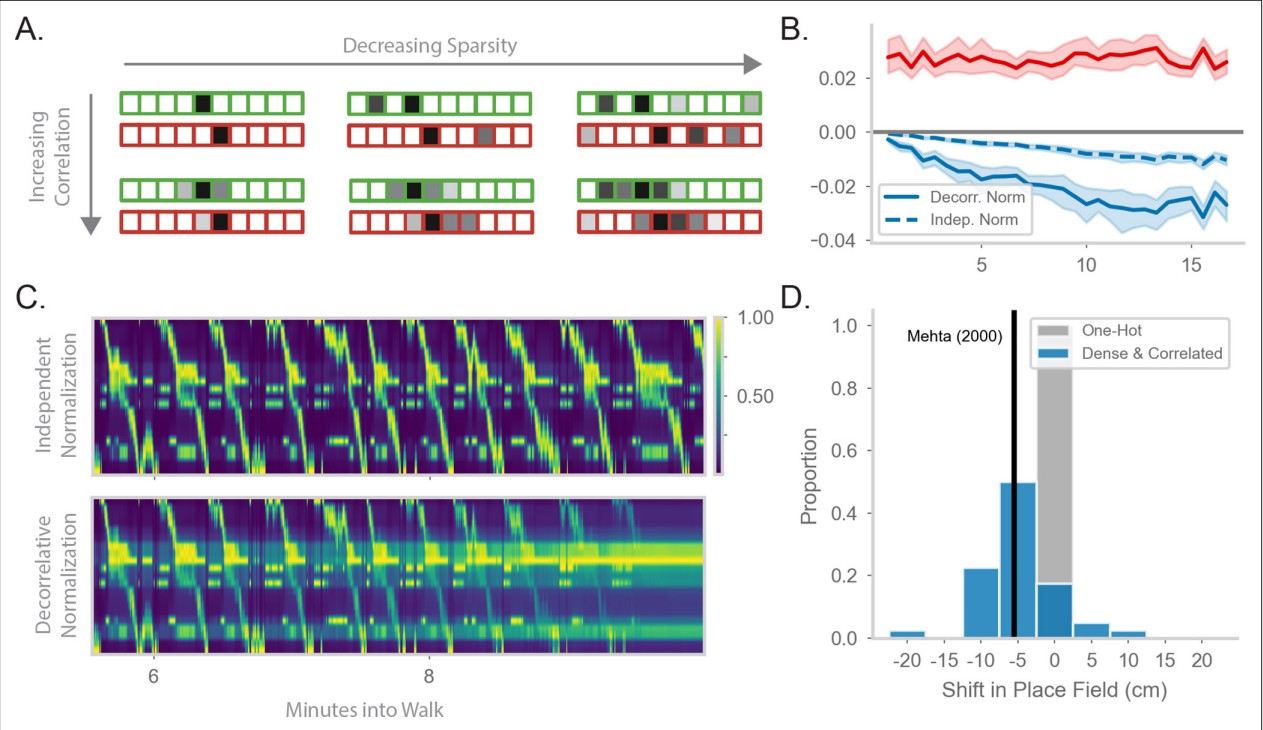

**Figure 4.** Generalizing the model to more realistic inputs. (**A**) Illustration of possible feature encodings $\phi$ for two spatially adjacent states in green and red. Feature encodings may vary in sparsity level and spatial correlation. (**B**) Average value of the STDP component (red) and the decorrelative normalization (solid blue) component of the gradient update over the course of a random walk. In dashed blue is a simpler Oja-like independent normalization update for comparison. Twenty-five simulations of forward-biased walks on a circular track were run, and shading shows 95% confidence interval. Input features are 3% sparse, with 10 cm spatial correlation. (**C**) Top: Example population activity of neurons in the RNN-S using the full decorrelative normalization rule over a 2min window of a forward-biased random walk. Population activity is normalized by the maximum firing rate. Bottom: As above, but for RNN-S using the simplified normalization update. (**D**) Shifts in place field peaks after a half hour simulation from the first two minutes of a 1D walk. Proportion of shifts in RNN-S with one-hot inputs shown in gray. Proportion of shifts in RNN-S with feature encodings (10% sparsity, 7.5 cm spatial correlation, $\gamma_R = 0.8$) shown in blue. Each data point is the average shift observed in one simulated walk, and each histogram is over 40 simulated walks. Solid line indicates the reported measure from *Mehta et al., 2000*.

The online version of this article includes the following figure supplement(s) for figure 4:

**Figure supplement 1.** Comparing place field shift and skew effects for different feature encodings.

without any changes to the synaptic weights. We show the effectiveness of separate network phases by simulating a 1D walk where the learning phase uses a small $\gamma_B$ (*Figure 3E*). Halfway through the walk, the animal enters a retrieval mode and accurately computes the SR with higher $\gamma_R$ (*Figure 3E*).

Under this scheme, the model can compute the SR for any $\gamma < 1$ (*Figure 3—figure supplement 1*). The separation of learning and retrieval phases stabilizes neural dynamics and allows flexible tuning of predictive power depending on task context.

## RNN-S can be generalized to more complex inputs with successor features

We wondered how RNN-S performs given more biologically realistic inputs. We have so far assumed that an external process has discretized the environment into uncorrelated states so that each possible state is represented by a unique input neuron. In other words, the inputs $\phi$ are one-hot vectors. However, inputs into the hippocampus are expected to be continuous and heterogeneous, with states encoded by overlapping sets of neurons (*Hardcastle et al., 2017*). When inputs are not one-hot, there is not always a canonical ground-truth $T$ matrix to fit and the predictive representations are referred to as successor features (*Barreto et al., 2017*; *Kulkarni et al., 2016*). In this setting, the performance of a model estimating successor features is evaluated by the temporal difference (TD) loss function.

Using the RNN-S model and update rule (*Equation 4*), we explore more realistic inputs $\phi$ and refer to $\phi$ as 'input features' for consistency with the successor feature literature. We vary the sparsity and

spatial correlation of the input features (*Figure 4A*). As before (*Figure 3H*), the network will operate in separate learning and retrieval modes, where $\gamma_B$ is below the critical value for stability. Under these conditions, the update rule will learn

$$J = R_{\phi\phi}(-1)R_{\phi\phi}(0)^{-1} \tag{7}$$

at steady state, where $R_{\phi\phi}(\tau)$ is the correlation matrix of $\phi$ with time lag $\tau$ (Appendix 3). Thus, the RNN-S update rule has the effect of normalizing the input feature via a decorrelative factor ($R_{\phi\phi}(0)^{-1}$) and mapping the normalized input to the feature expected at the next time step in a STDP-like manner ($R_{\phi\phi}(-1)$). This interpretation generalizes the result that $J = T^{\mathsf{T}}$ in the one-hot encoding case (Appendix 3).

We wanted to further explore the function of the normalization term. In the one-hot case, it operates over each synapse independently and makes a probability distribution. With more realistic inputs, it operates over a set of synapses and has a decorrelative effect. We first ask how the decorrelative term changes over learning of realistic inputs. We compare the mean value of the STDP term of the update ($x_i(t)x_j(t-1)$) to the normalization term of the update ($x_j(t-1)\sum_k J_{ik}x_k(t-1)$) during a sample walk (*Figure 4B*). The RNN-S learning rule has stronger potentiating effects in the beginning of the walk. As the model learns more of the environment and converges on the correct transition structure, the strength of the normalization term balances out the potentiation term. It may be that the normalization term is particularly important in maintaining this balance as inputs become more densely encoded. We test this hypothesis by using a normalization term that operates on each synapse independently (similar to Oja's Rule, *Oja, 1982*, Appendix 5). We see that the equilibrium between potentiating and depressing effects is not achieved by this type of independent normalization (*Figure 4B*, Appendix 6).

We wondered whether the decorrelative normalization term is necessary for the RNN-S to develop accurate representations. By replacing the decorrelative term with an independent normalization, features from non-adjacent states begin to be associated together and the model activity becomes spatially non-specific over time (*Figure 4C*, top). In contrast, using the decorrelative term, the RNN-S population activity is more localized (*Figure 4C*, bottom).

Interestingly, we noticed an additional feature of place maps as we transitioned from one-hot feature encodings to more complex feature encodings. We compared the representations learned by the RNN-S in a circular track walk with one-hot features versus more densely encoded features. For both input distributions, the RNN-S displayed the same skewing in place fields seen in *Figure 2* (*Figure 4—figure supplement 1*). However, the place field peaks of the RNN-S model additionally shifted backwards in space for the more complex feature encodings (*Figure 4D*). This was not seen for the one-hot encodings (*Figure 4D*). The shifting in the RNN-S model is consistent with the observations made in *Mehta et al., 2000* and demonstrates the utility of considering more complex input conditions. A similar observation was made in *Stachenfeld et al., 2017* with noisy state inputs. In both cases, field shifts could be caused by neurons receiving external inputs at more than one state, particularly at states leading up to its original field location.

## RNN-S estimates successor features even with naturalistic trajectories

We ask whether RNN-S can accurately estimate successor features, particularly under conditions of natural behavior. Specifically, we used the dataset from *Payne et al., 2021*, gathered from foraging Tufted Titmice in a 2D arena (*Figure 5A*). We discretize the arena into a set of states and encode each state as in Section 2.5. Using position-tracking data from *Payne et al., 2021*, we simulate the behavioral trajectory of the animal as transitions through the discrete state space. The inputs into the successor feature model are the features associated with the states in the behavioral trajectory.

We first wanted to test whether the RNN-S model was robust across a range of different types of input features. We calculate the TD loss of the model as a function of the spatial correlation across inputs $\phi$ (*Figure 5B*). We find that the model performs well across a range of inputs but loss is higher when inputs are spatially uncorrelated. This is consistent with the observation that behavioral transitions are spatially local, such that correlations across spatially adjacent features aid in the predictive power of the model. We next examine the model performance as a function of the sparsity of inputs $\phi$ (*Figure 5C*). We find the model also performs well across a range of feature sparsity, with lowest loss when features are sparse.

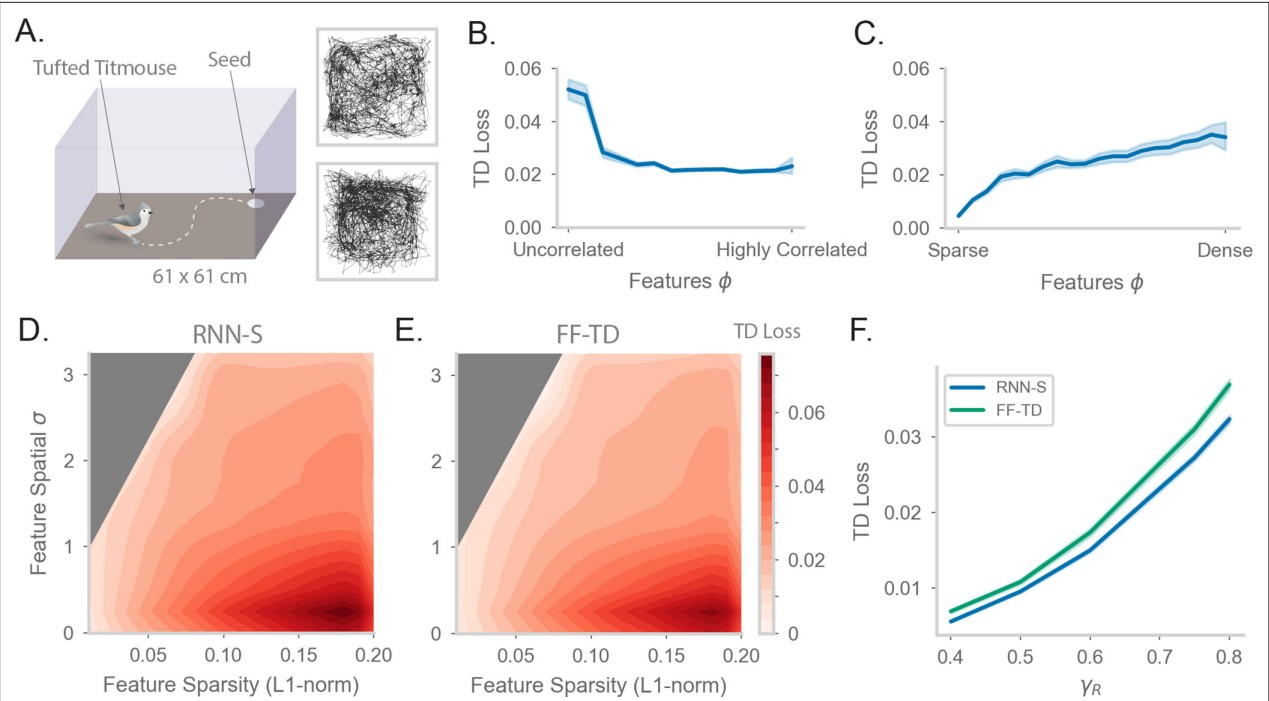

**Figure 5.** Fitting successor features to data with RNN-S over a variety of feature encodings. (**A**) We use behavioral data from Payne et al, where a Tufted Titmouse randomly forages in a 2D environment while electrophysiological data is collected (replicated with permission from authors). Two example trajectories are shown on the right. (**B**) Temporal difference (TD) loss versus the spatial correlation of the input dataset, aggregated over all sparsity levels. Here, $\gamma_R = 0.75$. Line shows mean, and shading shows 95% confidence interval. (**C**) As in (**B**), but measuring TD loss versus the sparsity level of the input dataset, aggregated over all spatial correlation levels. (**D**) TD loss for RNN-S with datasets with different spatial correlations and sparsities. Gray areas were not represented in the input dataset due to the feature generation process. Here, $\gamma_R = 0.75$, and three simulations were run for each spatial correlation and sparsity pairing under each chosen $\gamma_R$. (**E**) As in (**G**), but for FF-TD. (**F**) TD loss of each model as a function of $\gamma_R$, aggregated over all input encodings. Line shows mean, and shading shows 95% confidence interval.

The online version of this article includes the following figure supplement(s) for figure 5:

**Figure supplement 1.** Parameter sweep details and extended TD error plots.

To understand the interacting effects of spatial correlation and feature sparsity in more detail, we performed a parameter sweep over both of these parameters (*Figure 5D*, *Figure 5—figure supplement 1A-F*). We generated random patterns according to the desired sparsity and smoothness with a spatial filter to generate correlations. This means that the entire parameter space is not covered in our sweep (e.g. the top-left area with high correlation and high sparsity is not explored). Note that since we generate $\phi$ by randomly drawing patterns, the special case of one-hot encoding is also not included in the parameter sweep (one-hot encoding is already explored in *Figure 2*). The RNN-S seems to perform well across a wide range, with highest loss in regions of low spatial correlation and low sparsity.

We want to compare the TD loss of RNN-S to that of a non-biological model designed to minimize TD loss. We repeat the same parameter sweep over input features with the FF-TD model (*Figure 5E*, *Figure 5—figure supplement 1G*). The FF-TD model performs similarly to the RNN-S model, with lower TD loss in regions with low sparsity or higher correlation. We also tested how the performance of both models is affected by the strength of $\gamma_R$ (*Figure 5F*). Both models show a similar increase in TD loss as $\gamma_R$ increases, although the RNN-S has a slightly lower TD loss at high $\gamma$ than the FF-TD model. Both models perform substantially better than a random network with weights of comparable magnitude (*Figure 5—figure supplement 1D*).

Unlike in the one-hot case, there is no ground-truth $T$ matrix for non-one-hot inputs, so representations generated by RNN-S and FF-TD may look different, even at the same TD loss. Therefore, to compare the two models, it is important to compare representations to neural data.

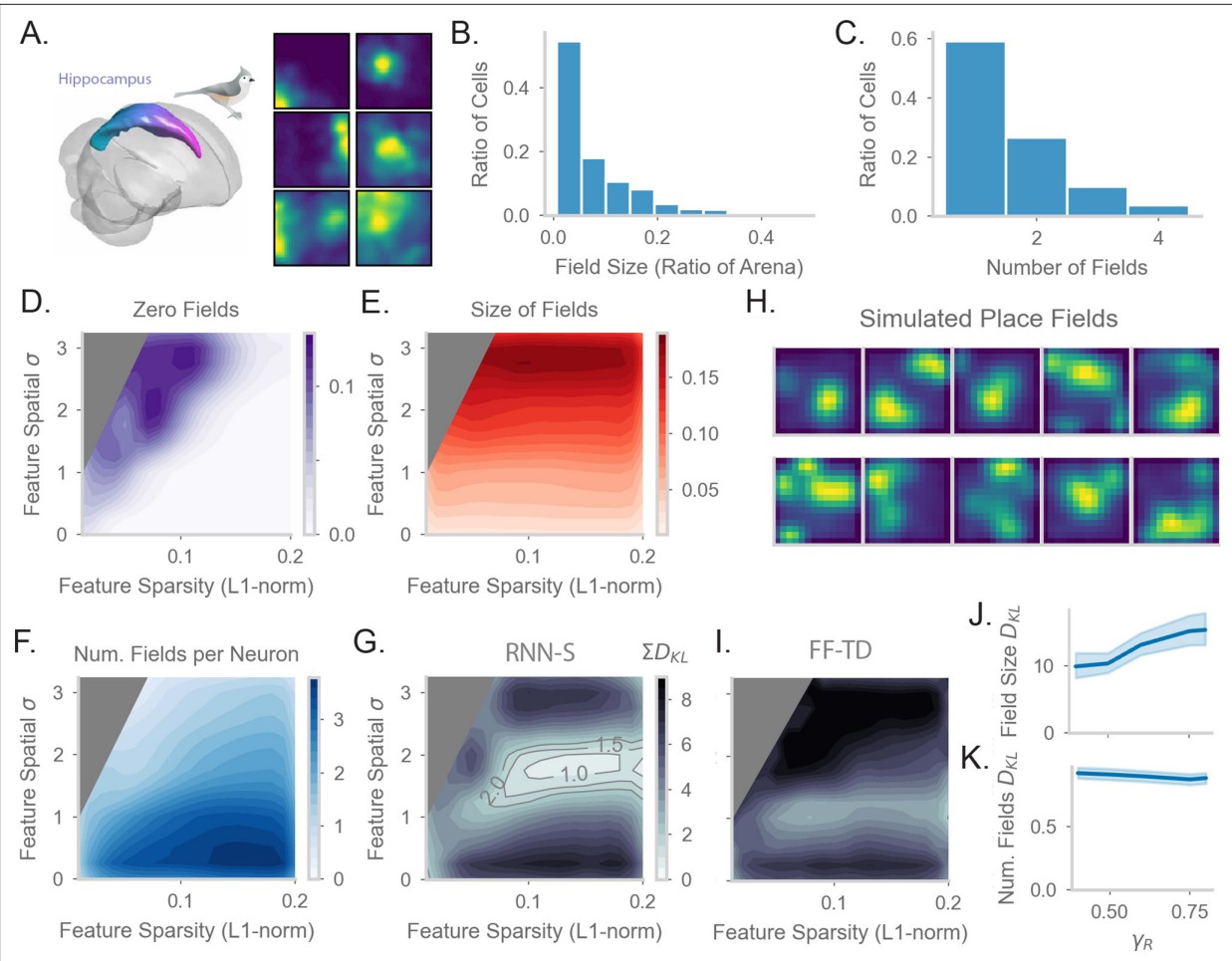

**Figure 6.** Comparing place fields from RNN-S to data. (**A**) Dataset is from Payne et al, where a Tufted Titmouse randomly forages in a 2D environment while electrophysiological data is collected (replicated with permission from authors). (**B**) Distribution of place cells with some number of fields, aggregated over all cells recorded in all birds. (**C**) Distribution of place cells with some field size as a ratio of the size of the arena, aggregated over all cells recorded in all birds. (**D**) Average proportion of non-place cells in RNN-S, aggregated over simulations of randomly drawn trajectories from Payne et al. Feature encodings are varied by spatial correlation and sparsity as in *Figure 5*. Each simulation used 196 neurons. As before, three simulations were run for each spatial correlation and sparsity pairing under each chosen $\gamma_R$. (**E**) As in (**D**), but for average field size of place cells. (**F**) As in (**D**), but for average number of fields per place cell. (**G**) As in (**D**) and (**E**), but comparing place cell statistics using the KL divergence ($D_{KL}$) between RNN-S and data from Payne et al. At each combination of input spatial correlation and sparsity, the distribution of field sizes is compared to the neural data, as is the distribution of number of fields per neuron, then the two $D_{KL}$ values are summed. Contour lines are drawn at $D_{KL}$ values of 1, 1.5, and 2 bits. (**H**) Place fields of cells chosen from the region of lowest KL divergence. (**I**) As in (**G**) but for FF-TD. (**J**) Change in KL divergence for field size as function of $\gamma$. Line shows mean, and shading shows 95% confidence interval. (**K**) Same as (**J**), but for number of fields.

The online version of this article includes the following figure supplement(s) for figure 6:

**Figure supplement 1.** Extended place field evaluation plots.

## RNN-S fits neural data in a random foraging task

Finally, we tested whether the neural representations learned by the models with behavioral trajectories from *Figure 5* match hippocampal firing patterns. We performed new analysis on neural data from *Payne et al., 2021* to establish a dataset for comparison. The neural data from *Payne et al., 2021* was collected from electrophysiological recordings in titmouse hippocampus during freely foraging behavior (*Figure 6A*). *Payne et al., 2021* discovered the presence of place cells in this area. We analyzed statistics of place cells recorded in the anterior region of the hippocampus, where homology with rodent dorsal hippocampus is hypothesized (*Tosches et al., 2018*). We calculated the distribution of place field size measured relative to the arena size (*Figure 6B*), as well as the distribution of the number of place fields per place cell (*Figure 6C*). Interestingly, with similar analysis methods,

*Henriksen et al., 2010* see similar statistics in the proximal region of dorsal CA1 in rats, indicating that our analyses could be applicable across organisms.

In order to test how spatial representations in the RNN-S are impacted by input features, we performed parameter sweeps over input statistics. As in *Payne et al., 2021*, we define place cells in the model as cells with at least one statistically significant place field under permutation tests. Under most of the parameter range, all RNN-S neurons would be identified as a place cell (*Figure 6D*). However, under conditions of high spatial correlation and low sparsity, a portion of neurons (12%) do not have any fields in the environment. These cells are excluded from further analysis. We measured how the size of place fields varies across the parameter range (*Figure 6E*). The size of the fields increases as a function of the spatial correlation of the inputs, but is relatively insensitive to sparsity. This effect can be explained as the spatial correlation of the inputs introducing an additional spatial spread in the neural activity. Similarly, we measured how the number of place fields per cell varies across the parameter range (*Figure 6F*). The number of fields is maximal for conditions in which input features are densely encoded and spatial correlation is low. These are conditions in which each neuron receives inputs from multiple, spatially distant states.

Finally, we wanted to identify regions of parameter space that were similar to the data of *Payne et al., 2021*. We measured the KL divergence between our model's place field statistics (*Figure 6DE*) and the statistics measured in *Payne et al., 2021* (*Figure 6BC*). We combined the KL divergence of both these distributions to find the parameter range in which the RNN-S best fits neural data (*Figure 6G*). This optimal parameter range occurs when inputs have a spatial correlation of $\sigma \approx 8.75$ cm and sparsity $\approx 0.15$. We note that the split-half noise floor of the dataset of *Payne et al., 2021* is a KL divergence of 0.12 bits (*Figure 6—figure supplement 1E*). We can visually confirm that the model fits the data well by plotting the place fields of RNN-S neurons (*Figure 6H*).

We wondered whether the predictive gain ($\gamma_R$) of the representations affects the ability of the RNN-S to fit data. The KL divergence changes only slightly as a function of $\gamma_R$. Mainly, the KL-divergence of the place field size increases as $\gamma_R$ increases (*Figure 6I*), but little effect is seen in the distribution of the number of place fields per neuron (*Figure 6J*).

We next tested whether the neural data was better fit by representations generated by RNN-S or the FF-TD model. Across all parameters of the input features, despite having similar TD loss (*Figure 5DE*), the FF-TD model has much higher divergence from neural data (*Figure 6GI*, *Figure 6—figure supplement 1*), similar to a random feedforward network (*Figure 6—figure supplement 1E*).

Overall, our RNN-S model seems to strike a balance between performance in estimating successor features, similarity to data, and biological plausibility. Furthermore, our analyses provide a prediction of the input structure into the hippocampus that is otherwise not evident in an algorithmic description or in a model that only considers one-hot feature encodings.

## Discussion

Hippocampal memory is thought to support a wide range of cognitive processes, especially those that involve forming associations or making predictions. However, the neural mechanisms that underlie these computations in the hippocampus are not fully understood. A promising biological substrate is the recurrent architecture of the CA3 region of the hippocampus and the plasticity rules observed. Here, we showed how a recurrent network with local learning rules can implement the successor representation, a predictive algorithm that captures many observations of hippocampal activity. We used our neural circuit model to make specific predictions of biological processes in this region.

A key component of our plasticity rule is a decorrelative term that depresses synapses based on coincident activity. Such anti-Hebbian or inhibitory effects are hypothesized to be broadly useful for learning, especially in unsupervised learning with overlapping input features (*Litwin-Kumar and Doiron, 2014*; *Sadeh and Clopath, 2021*; *Pehlevan et al., 2018*; *Payne et al., 2021*). Consistent with this hypothesis, anti-Hebbian learning has been implicated in circuits that perform a wide range of computations, from distinguishing patterns (*Földiák, 1990*), to familiarity detection (*Tyulmankov et al., 2022*), to learning birdsong syllables (*Mackevicius et al., 2020*). This inhibitory learning may be useful because it decorrelates redundant information, allowing for greater specificity and capacity in a network (*Sadeh and Clopath, 2021*; *Földiák, 1990*). Our results provide further support of these hypotheses and predict that anti-Hebbian learning is fundamental to a predictive neural circuit.

We derive an adaptive learning rate that allows our model to quickly learn a probability distribution, and generally adds flexibility to the learning process. The adaptive learning rate changes such that neurons that are more recently active have a slower learning rate. This is consistent with experimental findings of metaplasticity at synapses (*Abraham and Bear, 1996*; *Abraham, 2008*; *Hulme et al., 2014*), and theoretical proposals that metaplasticity tracks the uncertainty of information (*Aitchison et al., 2021*). In RNN-S, the adaptive learning rate improves the speed of learning and better recapitulates hippocampal data. Our adaptive learning rate also has interesting implications for flexible learning. Memory systems must be able to quickly learn new associations throughout their lifetime without catastrophe. Our learning rate is parameterized by a forgetting term $\lambda$ that controls the timescale in which environmental statistics are expected to be stationary. Although we fixed $\lambda = 1$ in our simulations, there are computational benefits in considering cases where $\lambda < 1$. This parameter provides a natural way for a memory system to forget gradually over time and prioritize recent experiences, in line with other theoretical studies that have also suggested that learning and forgetting on multiple timescales allow for more flexible behavior (*Kaplanis et al., 2018*; *Fusi et al., 2007*).

We tested the sensitivity of our network to various parameters and found a broad range of valid solutions. Prior work has sought to understand how an emergent property of a network could be generated by multiple unique solutions (*Goldman et al., 2001*; *Prinz et al., 2004*; *Bittner et al., 2021*; *Hertäg and Clopath, 2022*). It has been suggested that redundancy in solution space makes systems more robust, accounting for margins of error in the natural world (*Marder and Goaillard, 2006*; *Marder and Taylor, 2011*). In a similar vein, our parameter sweep over plasticity kernels revealed that a sizeable variety of kernels give solutions that resemble the SR. Although our model was initially sensitive to the value of $\gamma$, we found that adding biological components, such as nonlinear dynamics and separate network modes, broadened the solution space of the network.

Several useful features arise from the fact that RNN-S learns the transition matrix $T$ directly, while separating out the prediction timescale, $\gamma$, as a global gain factor. It is important for animals to engage in different horizons of prediction depending on task or memory demands (*Mattar and Lengyel, 2022*; *Bellmund et al., 2020*). In RNN-S, changing the prediction time horizon is as simple as increasing or decreasing the global gain of the network. Mechanistically, this could be accomplished by a neuromodulatory gain factor that boosts $\gamma$, perhaps by increasing the excitability of all neurons (*Heckman et al., 2009*; *Nadim and Bucher, 2014*). In RNN-S, it was useful to have low network gain during learning ($\gamma_B$), while allowing higher gain during retrieval to make longer timescale predictions ($\gamma_R$). This could be accomplished by a neuromodulatory factor that switches the network into a learning regime (*Pawlak et al., 2010*; *Brzosko et al., 2019*), for example acetylcholine, which reduces the gain of recurrent connections and increases learning rates (*Hasselmo, 1999*; *Hasselmo, 2006*). The idea that the hippocampus might compute the SR with flexible $\gamma$ could help reconcile recent results that hippocampal activity does not always match high-$\gamma$ SR (*Widloski and Foster, 2022*; *Duvelle et al., 2021*). Additionally, flexibility in predictive horizons could explain the different timescales of prediction observed across the anatomical axes of the hippocampus and entorhinal cortex (*Jung et al., 1994*; *Dolorfo and Amaral, 1998*; *Brun et al., 2008*; *Kjelstrup et al., 2008*; *Strange et al., 2014*; *Poppenk et al., 2013*; *Brunec and Momennejad, 2022*). Specifically, a series of successor networks with different values of $\gamma$ used in retrieval could establish a gradient of predictive timescales. Functionally, this may allow for learning hierarchies of state structure and could be useful for hierarchical planning (*McKenzie et al., 2014*; *Momennejad and Howard, 2018*; *Ribas-Fernandes et al., 2019*).

Estimating $T$ directly provides RNN-S with a means to sample likely future trajectories, or distributions of trajectories, which is computationally useful for many memory-guided cognitive tasks beyond reinforcement learning, including reasoning and inference (*Ostojic and Fusi, 2013*; *Goodman et al., 2016*). The representation afforded by $T$ may also be particularly accessible in neural circuits. *Ostojic and Fusi, 2013* note that only few general assumptions are needed for synaptic plasticity rules to estimate transition statistics. Thus, it is reasonable to assume that some form of transition statistics are encoded broadly across the brain.

Interestingly, we also found that the recurrent network fit hippocampal data better than a feedforward network. An interesting direction for further work involves untangling which brain areas and cognitive functions can be explained by deep (feed forward) neural networks (*Bonnen et al., 2021*), and which rely on recurrent architectures, or even richer combinations of generative structures (*Das et al., 2021*). Recurrent networks, such as RNN-S, support generative sequential sampling, reminiscent

of hippocampal replay, which has been proposed as a substrate for planning, imagination, and structural inference (*Foster and Wilson, 2006*; *Singer et al., 2013*; *Momennejad and Howard, 2018*; *Evans and Burgess, 2020*; *Kay et al., 2020*).

There are inherent limitations to the approach of using a recurrent network to estimate the SR. For instance, network dynamics can be prone to issues of instability due to the recurrent buildup of activity. To prevent this instability, we introduce two different modes of operation, 'learning' and 'retrieval'. An additional limitation is that errors in the estimated one-step transition can propagate over the course of the predictive rollout. This is especially problematic if features are more densely coded or more correlated, which makes one-step transition estimations more difficult. These insights into vulnerabilities of a recurrent network have interesting parallels in biology. Some hippocampal subfields are known to be highly recurrent (*Schaffer, 1892*; *Ramón and Cajal, 1904*; *Miles and Wong, 1986*; *Le Duigou et al., 2014*). This recurrency has been linked to the propensity of the hippocampus to enter unstable regimes, such as those that produce seizures (*Sparks et al., 2020*; *Thom, 2014*; *Lothman et al., 1991*; *Knight et al., 2012*). It remains an open question how a healthy hippocampus maintains stable activity, and to what extent the findings in models such as ours can suggest biological avenues to tame instability.

Other recent theoretical works have also sought to find biological mechanisms to learn successor representations, albeit with different approaches (*Brea et al., 2016*; *de Cothi and Barry, 2020*; *Bono et al., 2023*; *Lee, 2022*; *George et al., 2023*). For instance, the model from *George et al., 2023* explores a feedforward network that takes advantage of theta phase-precession to learn the SR. They analyze how place cells deform around boundaries and the function of the dorsal-ventral gradient in field size. The model introduced by *Bono et al., 2023* uses a feedforward network with hippocampal replay. They explore how replay can modulate the bias-variance tradeoff of their SR estimate and apply their model to fear-conditioning data. It is important to note that these mechanisms are not mutually exclusive with RNN-S. Taken together with our work, these models suggest that there are multiple ways to learn the SR in a biological circuit and that these representations may be more accessible to neural circuits than previously thought.

## Methods
### Code availability
Code is posted on Github: https://github.com/chingf/sr-project; *Fang, 2022*.

### Random walk simulations
We simulated random walks in 1D (circular track) and 2D (square) arenas. In 1D simulations, we varied the probability of staying in the current state and transitioning forwards or backwards to test different types of biases on top of a purely random walk. In 2D simulations, the probabilities of each possible action were equal. In our simulations, one timestep corresponds to 1/3 second and spatial bins are assumed to be 5 cm apart. This speed of movement (15 cm/s) was chosen to be consistent with previous experiments. In theory, one can imagine different choices of timestep size to access different time horizons of prediction– that is, the choice of timestep interacts with the choice of $\gamma$ in determining the prediction horizon.

### RNN-S model
This section provides details and pseudocode of the RNN-S simulation. Below are explanations of the most relevant variables:

| | |
|---|---|
| $N$ | Number of states, also equal to the number of neurons in the RNN |
| $\boldsymbol{x}$ | N-length vector of RNN neural activity |
| $J$ | $(N \times N)$ synaptic weight matrix |
| $M$ | $(N \times N)$ SR matrix |
| $\phi$ | $N$-length input vector into network |

*Continued on next page*

*Continued*

| | |
|---|---|
| $b$ | binary variable indicating learning (0) or retrieval (1) mode |
| $\gamma_B$ | Value of $\gamma$ the network uses to calculate $M$ in learning mode |
| $\gamma_R$ | Value of $\gamma$ the network uses to calculate $M$ in retrieval mode |
| $n$ | Variable that tracks the activity of neurons integrated over time |
| $\lambda$ | Discount value the network uses to calculate $n$ |
| $\eta$ | Learning rate |

The RNN-S algorithm is as follows:

---

**Algorithm 1** RNN-S.

---

Inputs:
 $\phi(t)$ for $t \in 1, \dots, T$
 $b(t)$ for $t \in 1, \dots, T$
Initialize:
 $J \leftarrow \mathbf{0}_{N \times N}$
 $\boldsymbol{n} \leftarrow \mathbf{0}_N$
 $\boldsymbol{x}(t) \leftarrow \mathbf{0}_N$ for $t \in 1, \dots, T$
for $t \in 1, \dots, T$ do
 if $b(t) == 1$ then              // Retrieval Mode
  $M^\mathsf{T} \leftarrow (1 - \gamma_R J)^{-1}$
  $\boldsymbol{x}(t) \leftarrow M^\mathsf{T} \phi(t)$
 else                   // Learning Mode
  $M^\mathsf{T} \leftarrow (1 - \gamma_B J)^{-1}$
  $\boldsymbol{x}(t) \leftarrow M^\mathsf{T} \phi(t)$
  $\boldsymbol{n} \leftarrow \boldsymbol{x}(t) + \lambda \boldsymbol{n}$         // Learning rate update
  $\Delta J \leftarrow \boldsymbol{x}(t)\boldsymbol{x}(t-1)^\mathsf{T} - (J\boldsymbol{x}(t-1))\boldsymbol{x}(t-1)^\mathsf{T}$ // Calculate weight update
  $\eta = \frac{1}{n}$         // Get learning rates (elementwise inversion)
  $\eta = \mathtt{min}(\eta, 1.0)$       // Learning rates can't exceed 1.0
  $J_{ij} \leftarrow J_{ij} + \eta_j \Delta J_{ij}$      // Update synaptic weight matrix
 end if
end for
return $\boldsymbol{x}$

---

## RNN-S with plasticity kernels

We introduce additional kernel-related variables to the RNN-S model above that are optimized by an evolutionary algorithm (see following methods subsection for more details):

| | |
|---|---|
| $A_+$, $\tau_+$ | pre$\rightarrow$ post side of the plasticity kernel $K_+(t) = A_+ E^{-t/\tau_+}$ |
| $A_-$, $\tau_-$ | As above, but for the post$\rightarrow$ pre side |
| $\alpha_d$ | Scaling term to allow for different self-synapse updates |
| $\alpha_n$ | Scaling term to allow for different learning rate updates |

We also define the variable $t_k = 20$, which is the length of the temporal support for the plasticity kernel. The value of $t_k$ was chosen such that $e^{-t_k/\tau}$ was negligibly small for the range of $\tau$ we were interested in. The update algorithm is the same as in Algorithm 1, except lines 15-16 are replaced with the following:

---

Algorithm 2 **Plasticity kernel update**

---

$\boldsymbol{n} \leftarrow \alpha_n \boldsymbol{x} + \lambda \boldsymbol{n}$ // Learning rate update
$\boldsymbol{k}_+ \leftarrow A_+ \sum_{t'=0}^{t_k} \boldsymbol{x}(t - t') e^{-t'/\tau_+}$ // Convolution with plasticity kernel
$\boldsymbol{k}_- \leftarrow A_- \sum_{t'=0}^{t_k} \boldsymbol{x}(t - t') e^{-t'/\tau_-}$
$\Delta J_K \leftarrow \boldsymbol{x}(t) \boldsymbol{k}_+^\mathsf{T} + \boldsymbol{k}_- \boldsymbol{x}(t)^\mathsf{T}$ // Calculate contribution to update from plasticity kernel
$\Delta J_K[ii] \leftarrow \alpha_d \boldsymbol{x}(t) \boldsymbol{k}_+^\mathsf{T}$ // Updates to self-synapses use separate scaling
$\Delta J \leftarrow \Delta J_K - (J\boldsymbol{x})\boldsymbol{x}^\mathsf{T}$ // Calculate weight update

---

## Metalearning of RNN parameters

To learn parameters of the RNN-S model, we use covariance matrix adaptation evolution strategy (CMA-ES) to learn the parameters of the plasticity rule. The training data provided are walks simulated from a random distribution of 1D walks. Walks varied in the number of states, the transition statistics, and the number of timesteps simulated. The loss function was the mean-squared error (MSE) loss between the RNN $J$ matrix and the ideal estimated $T$ matrix at the end of the walk.

## RNN-S with truncated recurrent steps and nonlinearity

For the RNN-S model with $t_{max}$ recurrent steps, lines 10 and 13 in Algorithm 1 is replaced with $M^\mathsf{T} \leftarrow \sum_{t=0}^{t_{max}} \gamma^t J^t$.

For RNN-S with nonlinear dynamics, there is no closed form solution. So, we select a value for $t_{max}$ and replace lines 10 and 13 in Algorithm 1 with an iterative update for $t_{max}$ steps: $\Delta \boldsymbol{x} = -\boldsymbol{x} + \gamma J \tanh(\boldsymbol{x'}) + \phi$. We choose $t_{max}$ such that $\gamma_{max}^t < 10^{-4}$.

## RNN-S with successor features

We use a tanh nonlinearity as described above. For simplicity, we set $\gamma_B = 0$.

## RNN-S with independent normalization

As in Algorithm 1, but with the following in place of line 16

$$\Delta J_{ij} \leftarrow x_i(t) x_j(t - 1) - J_{ij} x_j(t - 1)^2 \tag{8}$$

## FF-TD Model

In all simulations of the FF-TD model, we use the temporal difference update. We perform a small grid search over the learning rate $\eta$ to minimize error (for SR, this is the MSE between the true $M$ and estimated $M$; for successor features, this is the temporal difference error). In the one-hot SR case, the temporal difference update given an observed transition from state $s$ to state $s'$ is:

$$\Delta M_{ji} = \begin{cases} \gamma M_{s'i} - M_{si} & \text{if } s = j \neq i \\ 1 + \gamma M_{s'i} - M_{si} & \text{if } s = j = i \\ 0 & \text{otherwise} \end{cases} \tag{9}$$

for all synapses $j \rightarrow i$. Given arbitrarily structured inputs (as in the successor feature case), the temporal difference update is:

$$\Delta M^\mathsf{T} = \eta \left( \phi + \gamma M \phi' - M \phi \right) \phi^\mathsf{T} \tag{10}$$

or, equivalently,

$$\Delta M_{ji} = \eta \left( \phi_i + \gamma \sum_k M_{ki} \phi'_k - \sum_k M_{ki} \phi_k \right) \phi_j \tag{11}$$

## Generation of feature encodings for successor feature models

For a walk with $N$ states, we created $N$-dimensional feature vectors for each state. We choose an initial sparsity probability $p$ and create feature vectors as random binary vectors with probability $p$ of being

'on'. The feature vectors were then blurred by a 2D Gaussian filter with variance $\sigma$ with 1 standard deviation of support. The blurred features were then min-subtracted and max-normalized. The sparsity of each feature vector was calculated as the L1 norm divided by $N$. The sparsity $s$ of the dataset then was the median of all the sparsity values computed from the feature vectors. To vary the spatial correlation of the dataset we need only vary $\sigma$. To vary the sparsity $s$ of the dataset we need to vary $p$, then measure the final $s$ after blurring with $\sigma$. Note that, at large $\sigma$, the lowest sparsity values in our parameter sweep were not possible to achieve.

## Measuring TD loss for successor feature models

We use the standard TD loss function (*Equation 18*). To measure TD loss, at the end of the walk we take a random sample of observed transition pairs $(\phi, \phi')$. We use these transitions as the dataset to evaluate the loss function.

## Analysis of place field statistics

We use the open source dataset from *Payne et al., 2021*. We select for excitatory cells in the anterior tip of the hippocampus. We then select for place cells using standard measures (significantly place-modulated and stable over the course of the experiment).

We determined place field boundaries with a permutation test as in *Payne et al., 2021*. We then calculated the number of fields per neuron and the field size as in *Henriksen et al., 2010*. The same analyses were conducted for simulated neural data from the RNN-S and FF-TD models.

## Behavioral simulation of Payne et al

We use behavioral tracking data from *Payne et al., 2021*. For each simulation, we randomly select an experiment and randomly sample a 28-min window from that experiment. If the arena coverage is less than 85% during the window, we redo the sampling until the coverage requirement is satisfied. We then downsample the behavioral data so that the frame rate is the same as our simulation (3 FPS). Then, we divide the arena into a $14 \times 14$ grid. We discretize the continuous X/Y location data into these states. This sequence of states makes up the behavioral transitions that the model simulates.

## Place field plots

From the models, we get the activity of each model neuron over time. We make firing field plots with the same smoothing parameters as *Payne et al., 2021*.

## Citation diversity statement

Systemic discriminatory practices have been identified in neuroscience citations, and a 'citation diversity statement' has been proposed as an intervention (*Dworkin et al., 2020*; *Zurn et al., 2020*). There is evidence that quantifying discriminatory practices can lead to systemic improvements in academic settings (*Hopkins, 2002*). Many forms of discrimination could lead to a paper being undercited, for example authors being less widely known or less respected due to discrimination related to gender, race, sexuality, disability status, or socioeconomic background. We manually estimated the number of male and female first and last authors that we cited, acknowledging that this quantification ignores many known forms of discrimination, and fails to account for nonbinary/intersex/trans folks. In our citations, first-last author pairs were 62% male-male, 19% female-male, 9% male-female, and 10% female-female, somewhat similar to base rates in our field (biaswatchneuro.com). To familiarize ourselves with the literature, we used databases intended to counteract discrimination (blackinneuro. com, anneslist.net, connectedpapers.com). The process of making this statement improved our paper, and encouraged us to adopt less biased practices in selecting what papers to read and cite in the future. We were somewhat surprised and disappointed at how low the number of female authors were, despite being a female-female team ourselves. Citation practices alone are not enough to correct the power imbalances endemic in academic practice *National Academies of Sciences, 2018* — this requires corrections to how concrete power and resources are distributed.

## Acknowledgements

This work was supported through NSF NeuroNex Award DBI-1707398, the Gatsby Charitable Foundation, the New York Stem Cell Foundation (Robertson Neuroscience Investigator Award), National

Institutes of Health (NIH Director's New Innovator Award (DP2-AG071918)), and the Arnold and Mabel Beckman Foundation (Beckman Young Investigator Award). CF received support from the NSF Graduate Research Fellowship Program. ELM received support from the Simons Society of Fellows. We thank Jack Lindsey and Tom George for comments on the manuscript, as well as Stefano Fusi, William de Cothi, Kimberly Stachenfeld, and Caswell Barry for helpful discussions.

## Additional information

### Funding

| Funder | Grant reference number | Author |
|---|---|---|
| National Science Foundation | NeuroNex Award DBI-1707398 | LF Abbott<br>Ching Fang<br>Emily L Mackevicius |
| Gatsby Charitable Foundation | | LF Abbott<br>Ching Fang<br>Emily L Mackevicius |
| New York Stem Cell Foundation | Robertson Neuroscience Investigator Award | Ching Fang<br>Dmitriy Aronov<br>Emily L Mackevicius |
| National Institutes of Health | NIH Director's New Innovator Award (DP2-AG071918) | Ching Fang<br>Dmitriy Aronov<br>Emily L Mackevicius |
| Arnold and Mabel Beckman Foundation | Beckman Young Investigator Award | Ching Fang<br>Dmitriy Aronov<br>Emily L Mackevicius |
| National Science Foundation | Graduate Research Fellowship Program | Ching Fang |
| Simons Foundation | Society of Fellows | Emily L Mackevicius |

The funders had no role in study design, data collection and interpretation, or the decision to submit the work for publication.

### Author contributions

Ching Fang, Conceptualization, Resources, Software, Formal analysis, Funding acquisition, Validation, Methodology, Writing – original draft, Writing – review and editing; Dmitriy Aronov, Conceptualization, Resources, Supervision, Funding acquisition, Visualization, Methodology, Writing – original draft, Writing – review and editing; LF Abbott, Conceptualization, Resources, Formal analysis, Supervision, Funding acquisition, Visualization, Methodology, Writing – original draft, Writing – review and editing; Emily L Mackevicius, Conceptualization, Resources, Software, Formal analysis, Supervision, Funding acquisition, Validation, Visualization, Methodology, Writing – original draft, Writing – review and editing

### Author ORCIDs

Ching Fang http://orcid.org/0000-0003-3653-0057
Dmitriy Aronov http://orcid.org/0000-0002-8277-5074
Emily L Mackevicius http://orcid.org/0000-0001-6593-4398

### Decision letter and Author response

Decision letter https://doi.org/10.7554/eLife.80680.sa1
Author response https://doi.org/10.7554/eLife.80680.sa2

## Additional files

### Supplementary files
• Transparent reporting form

## Data availability

The current manuscript is a computational study, so no data have been generated for this manuscript. Modelling code is publicly available on GitHub: https://github.com/chingf/sr-project (copy archived at swh:1:rev:43320e9b8c15927c67849f768d2a9bf17f68a0ea).

The following previously published dataset was used:

| Author(s) | Year | Dataset title | Dataset URL | Database and Identifier |
|---|---|---|---|---|
| Payne H, Lynch G, Aronov D | 2021 | Neural representations of space in the hippocampus of a food-caching bird | https://doi.org/10.5061/dryad.pg4f4qrp7 | Dryad Digital Repository, 10.5061/dryad.pg4f4qrp7 |

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

# Appendix 1

## Finding the conditions to retrieve from RNN steady-state activity

The successor representation is defined as

$$M = (I - \gamma T)^{-1} \tag{12}$$

where $T$ is the transition probability matrix such that $T_{ji} = P(s' = i | s = j)$ for current state $s$ and future state $s'$

For an RNN with connectivity $J$, activity $x$, input $\phi$, and gain $\gamma \in [0, 1]$, the (linear) discrete-time dynamics equation is (**Amarimber, 1972**)

$$\Delta x = -x(t) + \gamma J x(t) + \phi(t). \tag{13}$$

Furthermore, the steady state solution can be found by setting $\Delta x = 0$

$$x_{SS} = (I - \gamma J)^{-1} \phi \tag{14}$$

Assume that $J = T^T$ as a result of the network using some STDP-like learning rule where pre-post connections are potentiated. The transposition is due to notational differences from the RL literature, where the $ij$ th index typically concerns the direction from state $i$ to state $j$. This is a result of differences in RL and RNN conventions in which inputs are left-multiplied and right-multiplied, respectively. Let $\gamma$ be a neuromodulatory factor that is applied over the whole network (and, thus, does not need to be encoded in the synaptic weights). Then, the equivalence to **equation 12** becomes clear and our steady state solution can be written as:

$$x_{SS} = M^T \phi \tag{15}$$

This is consistent with the successor representation framework shown in **Stachenfeld et al., 2017**, where the columns of the $M$ matrix represent the firing fields of a neuron, and the rows of the $M$ matrix represent the network response to some input.

## Appendix 2

### Deriving the RNN-S learning rule from TD Error and showing the learning rule is valid under a stability condition

Transitions between states $(s, s')$ are observed as features $\phi(s), \phi(s')$ where $\phi$ is some function. For notational simplicity, we will write these observed feature transitions as $(\phi, \phi')$. A dataset $\mathcal{D}$ is comprised of these observed feature transitions over a behavioral trajectory. Successor features are typically learned by some function approximator $\psi(\phi; \theta)$ that is parameterized by $\theta$ and takes in the inputs $\phi$. The SF approximator, $\psi$, is learned by minimizing the temporal difference (TD) loss function (*Sutton and Barto, 2018*):

$$L(\theta) = \mathbb{E}\left[\left\|\phi + \gamma\psi^\pi(\phi'; \theta) - \psi(\phi; \theta)\right\|_2^2 | \mathcal{D}\right] \tag{16}$$

for the current policy $\pi$. Here, the TD target is $\phi + \gamma\psi^\pi(\phi'; \theta)$. Analogous to the model-free setting where the value function $V$ is being learned, $\phi$ is in place of the reward $r$. Following these definitions, we can view the RNN-S as the function approximator $\psi$:

$$\psi(\phi; \theta = J) = (I - \gamma J)^{-1}\phi \tag{17}$$

For a single transition $(\phi, \phi')$ we can write out the loss as follows:

$$L(\theta) = \left\|\phi + \gamma\psi^\pi(\phi'; \theta) - (I - \gamma J)^{-1}\phi\right\|_2^2 \tag{18}$$

For each observed transition, we would like to update $\psi$ such that the loss $L$ is minimized. Thus, we take the gradient of this temporal difference loss function with respect to our parameter $\theta = J$:

$$\nabla_J L(\theta) = 2\left(\phi + \gamma\psi^\pi(\phi'; \theta) - (I - \gamma J)^{-1}\phi\right)\nabla_J\left(-(I - \gamma J)^{-1}\phi\right)^\mathsf{T} \tag{19}$$

We can make the TD approximation (*Sutton and Barto, 2018*):

$$\nabla_J L(\theta) = 2\left(\phi + \gamma(I - \gamma J)^{-1}\phi' - (I - \gamma J)^{-1}\phi\right)\nabla_J\left(-(I - \gamma J)^{-1}\phi\right)^\mathsf{T} \tag{20}$$

$$= 2\left(\phi + \gamma(I - \gamma J)^{-1}\phi' - (I - \gamma J)^{-1}\phi\right)\left(-(I - \gamma J)^{-1}(-\gamma)(I - \gamma J)^{-1}\phi\right)^\mathsf{T} \tag{21}$$

$$= -2\left((I - \gamma J)x + \gamma x' - x\right)\left(\gamma(I - \gamma J)^{-1}x\right)^\mathsf{T} \tag{22}$$

$$= -2\gamma^2\left(x' - Jx\right)\left((I - \gamma J)^{-1}x\right)^\mathsf{T} \tag{23}$$

$$= -2\gamma^2(x' - Jx)x^\mathsf{T}(I - \gamma J)^{-\mathsf{T}} \tag{24}$$

While $-\nabla_J L(\theta)$ gives the direction of steepest descent in the loss, we will consider a linear transformation of the gradient that allows for a simpler update rule. This simpler update rule will be more amenable to a biologically plausible learning rule. We define this modified gradient as $D = \nabla_J L(\theta)M$ where $M = (I - \gamma J)^\mathsf{T}$. We must first understand the condition for $D$ to be in a direction of descent:

$$\langle D, \nabla_J L \rangle > 0 \tag{25}$$

$$\mathrm{Tr}(D^\mathsf{T}\nabla_J L) > 0 \tag{26}$$

$$\mathrm{Tr}(\nabla_J L M \nabla_J L) > 0 \tag{27}$$

$$\mathrm{Tr}\left(\nabla_J L\left(\frac{M + M^\mathsf{T}}{2} + \frac{M - M^\mathsf{T}}{2}\right)\nabla_J L\right) > 0 \tag{28}$$

$$\frac{1}{2} \operatorname{Tr}(\nabla_J L (M + M^\mathsf{T}) \nabla_J L) > 0 \tag{29}$$

This expression is satisfied if $M + M^\mathsf{T}$ is positive definite (its eigenvalues are positive). Thus, we find that our modified gradient points towards a descent direction if the eigenvalues of $M + M^\mathsf{T}$ are positive. Interestingly, this condition is equivalent to stating that the recurrent network dynamics are stable and do not exhibit non-normal amplification (*Kumar et al., 2022*; *Murphy and Miller, 2009*; *Goldman, 2009*). In other words, as long as the network dynamics are in a stable regime and do not have non-normal amplification, our modified gradient reduces the temporal difference loss. Otherwise, the gradient will not point towards a descent direction.

We will use the modified gradient $-D = (x' - Jx)x^\mathsf{T}$ as our synaptic weight update rule. Our theoretical analysis explains much of the results seen in the main text. As the gain parameter $\gamma_B$ is increased, the network is closer to the edge of stability (the eigenvalues of $M$ are close to positive values, *Figure 3A*). Stability itself is not enough to guarantee that our update rule is valid. We need the additional constraint that non-normal amplification should not be present (eigenvalues of $M + M^\mathsf{T}$ are positive). In practice, however, this does not seem to be a mode that affects our network. That is, the $\gamma_B$ value for which the error in the network increases coincides with the $\gamma_B$ value for which the network is no longer stable (*Figure 3B*). Our theoretical analysis also shows that the gain $\gamma_B$ can always be decreased such that the eigenvalues of $M + M^\mathsf{T}$ are positive and our update rule is valid (*Figure 3E*). At the most extreme, one can set $\gamma_B = 0$ during learning to maintain stability (as we do in *Figure 4* and onwards).

## Appendix 3

### Proving the RNN-S update rule calculated on firing rates ($x$) depends only on feedforward inputs ($\phi$) at steady state

We will show that our update rule, which uses $x$ (neural activity), converges on a solution that depends only on $\phi$ (the feedforward inputs). We will also show that in the one-hot case, we learn the SR exactly.

As a reminder, our learning rule for each $j \rightarrow i$ synapse is:

$$\Delta J = \eta(x' - Jx)x^\mathsf{T} \tag{30}$$

We can solve for the steady state solution of *Equation 30* (set $\Delta J = 0$). Let $A = (1 - \gamma J)^{-1}$ for notational convenience, and recall that in steady state $x = A\phi$. Let $\langle x \rangle$ denote the average of $x$ over time.

$$J = \langle x'x^\mathsf{T}\rangle\langle xx^\mathsf{T}\rangle^{-1} \tag{31}$$

$$J = \langle A\phi'(A\phi)^\mathsf{T}\rangle\langle A\phi(A\phi)^\mathsf{T}\rangle^{-1} \tag{32}$$

$$J = \langle A\phi'\phi^\mathsf{T}A^\mathsf{T}\rangle\langle A\phi\phi^\mathsf{T}A^\mathsf{T}\rangle^{-1} \tag{33}$$

$$J = A\langle\phi'\phi^\mathsf{T}\rangle A^\mathsf{T}(A\langle\phi\phi^\mathsf{T}\rangle A^\mathsf{T})^{-1} \tag{34}$$

Note that, since $A = (1 - \gamma J)^{-1}$, $J = \frac{1}{\gamma}(1 - A^{-1})$.

$$A\langle\phi'\phi^\mathsf{T}\rangle A^\mathsf{T} = \frac{1}{\gamma}(1 - A^{-1})A\langle\phi\phi^\mathsf{T}\rangle A^\mathsf{T} \tag{35}$$

$$A\langle\phi'\phi^\mathsf{T}\rangle A^\mathsf{T} = \frac{1}{\gamma}(A\langle\phi\phi^\mathsf{T}\rangle A^\mathsf{T} - \langle\phi\phi^\mathsf{T}\rangle A^\mathsf{T}) \tag{36}$$

$$A\langle\phi'\phi^\mathsf{T}\rangle = \frac{1}{\gamma}(A\langle\phi\phi^\mathsf{T}\rangle - \langle\phi\phi^\mathsf{T}\rangle) \tag{37}$$

Thus,

$$\langle\phi'\phi^\mathsf{T}\rangle = \frac{1}{\gamma}(1 - A^{-1})\langle\phi\phi^\mathsf{T}\rangle \tag{38}$$

Therefore,

$$J = \langle\phi'\phi^\mathsf{T}\rangle\langle\phi\phi^\mathsf{T}\rangle^{-1} \tag{39}$$

$$J = R_{\phi\phi}(-1)R_{\phi\phi}(0)^{-1} \tag{40}$$

where $R_{\phi\phi}(\tau)$ is the autocorrelation matrix for some time lag $\tau$. Therefore, the RNN-S weight matrix $J$ at steady state is only dependent on the inputs into the RNN over time.

In the case where $\phi$ is one-hot, we compute the SR exactly. This is because the steady state solution at each $j \rightarrow i$ synapse simplifies into the following expression:

$$J_{ij} = \frac{\sum_{t'} \phi_j(t' - 1)\phi_i(t')}{\sum_{t'} \phi_j(t')} \tag{41}$$

This is the definition of the transition probability matrix and we see that $J = T^\mathsf{T}$. Note that the solution for $J_{ij}$ in *Equation 41* is undefined if state $j$ is never visited. We assume each relevant state is visited at least once here.

## Appendix 4

### Deriving the adaptive learning rate update rule

This section explains how the adaptive learning rate is derived. The logic will be similar to calculating a weighted running average. Let $d_{ij}(t)$ be a binary function that is 1 if the transition from timestep $t-1$ to timestep $t$ is state $j$ to state $i$. Otherwise, it is 0. Assume $\phi$ is one-hot encoded. Notice that in the one-hot case, the RNN-S update rule (**Equation 4**) simplifies to:

$$\Delta J_{ij} \approx \eta(d_{ij} - J_{ij}x_j) \tag{42}$$

What $\eta$ should be used so $J$ approaches $T^\mathsf{T}$ as quickly as possible? During learning, the empirical transition matrix, $T(t)$, changes at each timestep $t$, based on transitions the animal has experienced. Define the total number of times that state $\phi_j$ happened prior to time $t$ as $n_j(t) = \sum_{t'=1}^{\infty} \phi_j(t-t')$, and define the running count of transitions from state j to state i as $c_{ij}(t) = \sum_{t'=1}^{\infty} d_{ij}(t-t')$. We want $J(t) = T^\mathsf{T}(t)$, which necessitates

$$\Delta J_{ij}(t) = T_{ji}(t) - T_{ji}(t-1) = \frac{c_{ij}(t)}{n_j(t)} - \frac{c_{ij}(t-1)}{n_j(t-1)} \tag{43}$$

$$= \frac{n_j(t-1)c_{ij}(t) - c_{ij}(t-1)n_j(t)}{n_j(t)n_j(t-1)} \tag{44}$$

Note that $n_j(t) = n_j(t-1) + \phi_j(t-1)$, and $c_{ij}(t) = c_{ij}(t-1) + d_{ij}(t)$, which gives us

$$\Delta J_{ij}(t) = \frac{n_j(t-1)c_{ij}(t-1) + n_j(t-1)d_{ij}(t) - c_{ij}(t-1)n_j(t-1) - c_{ij}(t-1)\phi_j(t-1)}{n_j(t)n_j(t-1)} \tag{45}$$

$$= \frac{n_j(t-1)d_{ij}(t) - c_{ij}(t-1)\phi_j(t-1)}{n_j(t)n_j(t-1)} \tag{46}$$

$$= \frac{1}{n_j(t)}\left(d_{ij}(t) - \frac{c_{ij}(t-1)\phi_j(t-1)}{n_j(t-1)}\right) \tag{47}$$

$$= \frac{1}{n_j(t)}\left(d_{ij}(t) - T_{ji}\phi_j(t-1)\right) \tag{48}$$

Therefore, comparing with **Equation 42**, we can see that a learning rate $\eta_j = \frac{1}{n_j(t)}$ will let $J = T^\mathsf{T}$ as quickly as possible. We have defined $n$ in terms of the inputs $\phi$ for this derivation, but in practice the adaptive learning rate as a function of $x$ works well with the RNN-S update rule (which is also a function of $x$). Thus, we use the adaptive learning rate defined over $x$ in our combined learning rule for increased biological plausibility.

In its current form, the update equation assumes transitions across all history of inputs are integrated. In reality, there is likely some kind of memory decay. This can be implemented with a decay term $\lambda \in (0, 1)$:

$$n_j(t) = \sum_{t'=1}^{\infty} \lambda^{t'} x_j(t-t') \tag{49}$$

$\lambda$ determines the recency bias over the observed transitions that make up the $T$ estimate. The addition of $\lambda$ has the added benefit that it naturally provides a mechanism for learning rates to modulate over time. If $\lambda = 1$, the learning rate can only monotonically decrease. If $\lambda < 1$, the learning rate can become strong again over time if a state has not been visited in a while. This provides a mechanism for fast learning of new associations, which is useful for a variety of effects, including remapping.

## Appendix 5

### Endotaxis model and the successor representation

The learning rule and architecture of our model is similar to a hypothesized "endotaxis" model (*Zhang et al., 2021*). In the endotaxis model, neurons fire most strongly near a reward, allowing the animal to navigate up a gradient of neural activity akin to navigating up an odor gradient. The endotaxis model discovers the structure of an environment and can solve many tasks such as spatial navigation and abstract puzzles. We were interested in similarities between RNN-S and the learning rules for endotaxis, in support of the idea that SR-like representations may be used by the brain for a broad range of intelligent behaviors. Here, we outline similarities and differences between the two model architectures.

The endotaxis paper (*Zhang et al., 2021*) uses Oja's rule in an RNN with place-like inputs. The SR can also be learned with an Oja-like learning rule. Oja's rule is typically written as (*Oja, 1982*):

$$\Delta J_{ij} = \eta x_j x_i - \eta J_{ij} x_i^2 \tag{50}$$

If we assume that there is a temporal asymmetry to the potentiation term (e.g., potentiation is more STDP-like than Hebbian), then we have

$$\Delta J_{ij} = \eta x_j(t-1)x_i(t) - \eta J_{ij} x_i(t)^2 \tag{51}$$

We then solve for the steady state solution of this equation, when $\Delta J_{ij} = 0$:

$$0 = \eta \langle x_j(t-1)x_i(t) \rangle - \eta J_{ij} \langle x_i(t)^2 \rangle \tag{52}$$

$$J_{ij} = \frac{\langle x_j(t-1)x_i(t) \rangle}{\langle x_i(t)^2 \rangle} \tag{53}$$

$$J_{ij} = \frac{\sum_{t'} x_j(t'-1)x_i(t')}{\sum_{t'} x_i(t')^2} \tag{54}$$

where $\langle \cdot \rangle$ indicates the time-average of some term. Assume that the plasticity rule does not use $x$ exactly, but instead uses $\phi$ directly. Given that inputs are one-hot encodings of the animal's state at some time $t$, the expression becomes

$$J_{ij} = \frac{\sum_{t'} \phi_j(t'-1)\phi_i(t')}{\sum_{t'} \phi_i(t')} \tag{55}$$

If we assume $T$ is symmetric, $J = T^\mathsf{T}$. Alternatively, if we use pre-synaptic normalization as opposed to the standard post-synaptic normalization of Oja's rule (i.e., index $j$ instead of $i$ in the denominator), we also have $J = T^\mathsf{T}$. Thus, the steady state activity of a RNN with this learning rule retrieves the SR, as shown in subsection 4.14.

# Appendix 6

## Independent normalization and successor features

If we assume the same Oja-like rule as in Appendix 5, we can also arrive at a similar interpretation in the successor feature case as in *Equation 7*. By solving for the steady state solution without any assumptions about the inputs $\phi$, we get the following equation:

$$J = R_{\phi\phi}(-1)\mathtt{diag}(R_{\phi\phi}(0))^{-1} \tag{56}$$

where diag is a function that retains only the diagonal of the matrix. This expression provides a useful way to contrast the learning rule used in RNN-S with an Oja-like alternative. While RNN-S normalizes by the full autocorrelation matrix, an Oja-like rule only normalizes by the diagonal of the matrix. This is the basis of our independent normalization model in *Figure 4BC*.

## Appendix 7

### Comparing alternate forms of normalizing the synaptic weight matrix

The anti-Hebbian term of the RNN-S learning rule normalizes the synaptic weight matrix into exactly a transition probability matrix. We wanted to test how important it was to use this exact normalization and whether other forms of the synaptic weight matrix could yield similar results. We simulated representations that would arise from different normalization procedures. For these tests, we simulate a random walk on a circular track, as in *Figure 2*, for 10 minutes of simulation time. A model where the synaptic weight matrix exactly estimates the transition probability matrix (as in *Equation 4*) will give the SR matrix (*Appendix 7—figure 1A*).

We test a model where the normalization term for the synaptic weight matrix is removed. Thus, $J$ will be equal to the count of observed transitions, i.e. $J_{ij}$ is equal to the number of experienced transitions from state $j$ to state $i$. We will refer to this as a count matrix. Without normalization, the values in the count matrix will increase steadily over the course of the simulation. This quickly results in unstable dynamics from the weights of the matrix being too large (*Appendix 7—figure 1B*). A simple way to prevent instability (specifically, ensure the maximum eigenvalue of the synaptic weight matrix is below 1) is to use an additional scaling factor $\alpha$ over the weights of the matrix, such that $J$ is multiplied by $\frac{1}{\alpha \max(J)}$. A careful choice of a scaling value can ensure network activity remains stable within this walk, although this is not a generalizable solution as different scaling values may be needed for different random walks and tasks. However, even with this modification the representations above are not sufficiently predictive compared to the original SR (the off-diagonal elements of the SR are not captured well), and the activity strength seems to be unevenly distributed across the states (*Appendix 7—figure 1CD*). It is likely that depressing all synapses by the same factor (similar to *Fiete et al., 2010*) does not correct for differences in occupancies. In other words, states that happen to be visited more by chance are likely to dominate the dynamics, even if the transition statistics are identical across all states.

Final, as a further test of different ways of parameterizing the synaptic weight matrix, we examine the steady state neural activity when the count matrix is instead scaled in a row-by-row fashion (*Appendix 7—figure 1E*). Specifically, we divide each row $i$ of the count matrix by the maximum of row $i$ (and some global scaling factor to ensure stability). Note that this is in contrast to $T$, where each row is divided by its sum. This is closer to the SR matrix expected if the synaptic weight matrix estimates $T$. We see there is a slight unevenness early on in learning in the diagonal of the matrix (*Appendix 7—figure 1E*). However, given enough observed transitions, the predictive representation looks reasonable and quite similar to the SR matrix.

Overall, we see that there are likely other formulations of the synaptic weight matrix that can give a representation similar to the SR. The important ingredient for this to happen appears to be some type of row-dependent normalization-- that is, neurons should have their synaptic weights normalized independently of each other. This ensures that occupancy is not conflated with transition statistics.

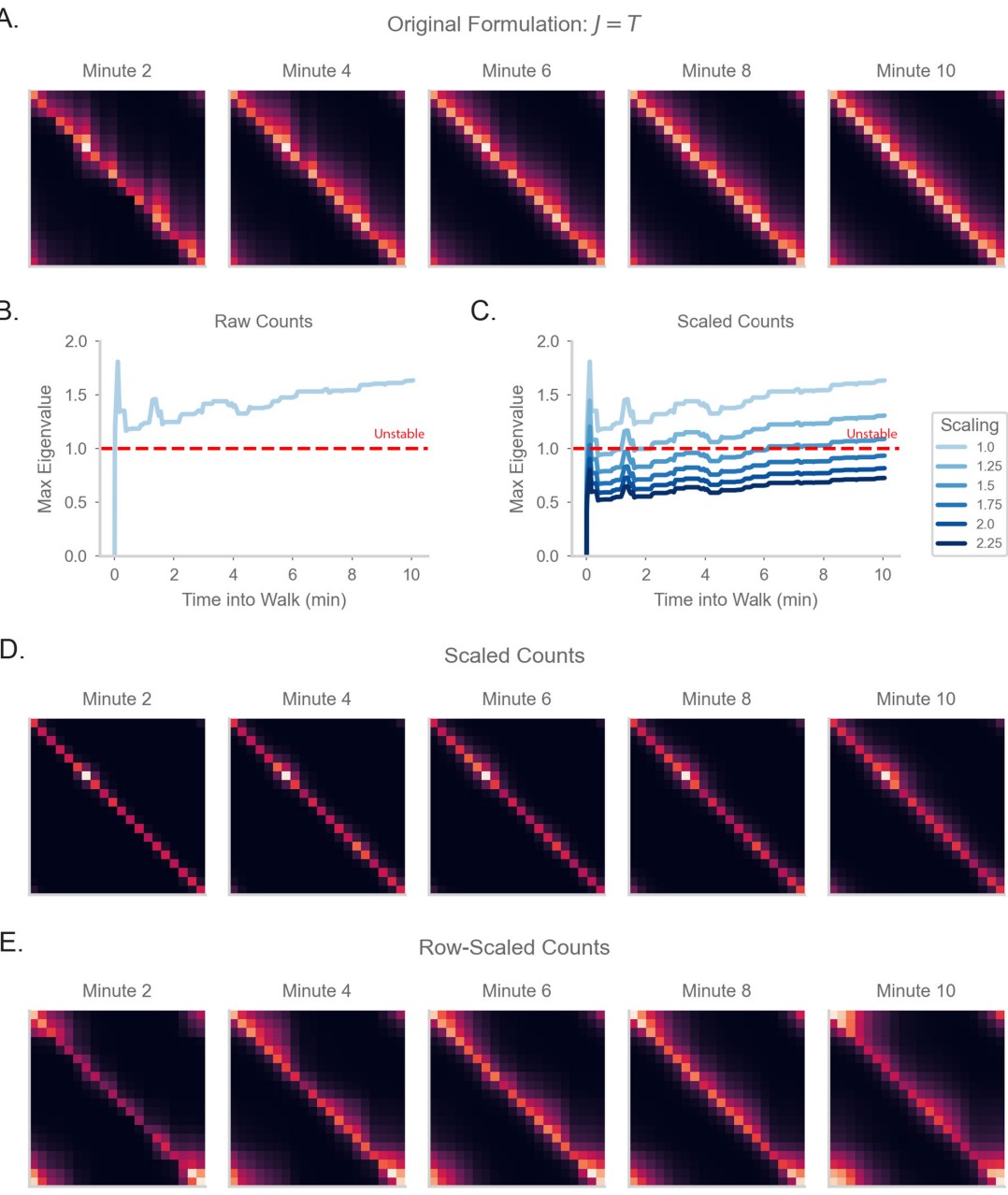

**Appendix 7—figure 1.** SR matrices under different forms of normalization. (**A**) The resulting SR matrix from a random walk on a circular track for 10minutes, if the synaptic weight matrix exactly estimates the transition probability matrix (as in *Equation 4*). (**B**) Model as in (**A**), but with normalization removed. Thus, $J$ will be equal to the count of observed transitions, i.e. $J_{ij}$ is equal to the number of experienced transitions from state $j$ to state $i$. We will refer to this as a count matrix. The plot show the maximum eigenvalue of the weight matrix, where an eigenvalue $\geq 1$ indicates instability (*Sompolinsky et al., 1988*). (**C**) As in (**B**), but with an additional scaling factor $\alpha$ over the weights of the matrix, such that $J$ is multiplied by $\frac{1}{\alpha \max(J)}$. (**D**) Steady state neural activity of the model in (**C**) with scaling factor 1.75. (**E**) As in (**D**), but the count matrix is instead scaled in a row-by-row fashion. Specifically, we divide each row $i$ of the count matrix by the maximum of row $i$ (and some global scaling factor to ensure stability).

