## [Editor Report]

This important work provides compelling evidence for the biological plausibility of the Successor Representation (SR) algorithm. The SR is a leading computational hypothesis to explore whether neural representations are consistent with the hypothesis that the neural networks in specific brain areas perform predictive computations. Establishing a biologically plausible learning rule for SR representations to form is of high significance in the field of neuroscience.

---

## [Decision Letter]

**Decision letter after peer review:**

Thank you for submitting your article "Neural learning rules for generating flexible predictions and computing the successor representation" for consideration by *eLife*. Your article has been reviewed by 3 peer reviewers, and the evaluation has been overseen by a Reviewing Editor and Timothy Behrens as the Senior Editor. The following individuals involved in review of your submission have agreed to reveal their identity: Stefano Recanatesi (Reviewer #1); Arthur Juliani (Reviewer #2); Srdjan Ostojic (Reviewer #3).

Main Comments:

1) The form of the plasticity rule in Equation 4 is motivated by the requirement that synaptic weights encode a properly normalised transition probability matrix (lines 92-96). But why is the normalisation important? What would change if synaptic weights were simply monotonic functions of transition probabilities, without normalisation? Presumably that would allow for a broader range of plasticity rules.

2) As the results of the paper strongly rely on the normalizing term in Equation 4. One of the reviewers suggests potentially moving upfront part of the discussion of this term, and enlarging the paragraph that discusses the biological plausibility of this specific term. Clearly laying out, for the non-expert reader, why it is biologically plausible compared to other learning rules. Also consider moving the required material to establish the novelty of such term: a targeted review of the relevant literature (current lines 358-366 and 413-433). This would allow the reader to understand immediately the significance and relative novelty of such term. For example, this reviewer personally wondered while reading the paper of how different was such term from the basic idea of Fiete et al., Neuron 2010 (DOI 10.1016/j.neuron.2010.02.003).

3) Related to the first point, the text insists on the fact that \γ is not encoded in the synaptic weights (eg line 89). Again, it is not entirely clear why this is important and justified, since γ is an ad-hoc factor in Equation 2. Presumably the proper normalisation of γ relies on the normalisation of J discussed above? It seems that this constraint could be relaxed.

4) As a consequence of the body of the text being devoted to the analysis of the design choices behind the proposed model, a relatively smaller portion of the work involves direct comparisons with neural data. In these comparisons, while it is apparent that there is a reasonable match between the proposed model and the empirical data, it is difficult to interpret these results. This is because it is unclear what should be expected of a good or bad model given the data being analyzed (TD error and KL divergence), and reasonable baselines to compare against are not presented outside of the traditional TD algorithm, which is shown to be comparable to the proposed RNN based method in a number of cases.

5) It would be useful to have a "limitations" paragraph in the discussion clearly outlining what this learning rule couldn't achieve. For example, Stachenfeld et al., Nat.Neuro. have many examples where the SR is deployed. Does the learning rule suggested by the authors would always work across the board, or are there limitations that could be highlighted where the framework suggested would not work well. No need to perform more experiments/simulations but simply to share insight regarding the results and the capability of the proposed learning rule.

Other comments/suggestions:

– Page 1: The introduction motivates this work with a discussion of hippocampal memory (storage and retrieval), but the work focuses on the SR which is inherently prospective. The first paragraph of the text could be revised to better make this connection beyond simply stating that the hippocampus is involved in both memory and future prediction.

– Page 2: The end of the introduction would be stronger if the motivation for an RNNs usage was tied to the literature on the known recurrent dynamics of the hippocampus. See for example: https://www.frontiersin.org/articles/10.3389/fncel.2013.00262/full

– Page 6: It is not clear the extent to which the FF-TD model differs from a canonical tabular SR algorithm or linear SF algorithm. My understanding is that it is the same, but the presentation in Figure 1i for example makes this somewhat unclear.

– Pages 6 – 11: it may be of benefit to more strongly support the various modifications to the model with connections to known or hypothesized hippocampal neural dynamics.

– Page 14: It states that "We discretize the arena into a set of states and encode each state as a randomly drawn feature ϕ." If I understand correctly, these features are not completely random, and instead follow the distribution described in Section 2.5. As it currently reads, it seems that these features might be drawn from a uniform random distribution, which would be misleading.

– Page 14: In Section 2.6 there is an assumption that a certain level of TD error corresponds to good performance. It is not clear what should objectively be considered a reasonable TD error. This is especially difficult to interpret in the case where both the RNN-S and FF-TD models display comparable performance. Is there perhaps some other baseline you would expect to perform considerably worse?

– Page 17: In Figure 4 it is somewhat confusing that the KL divergence (subplots G and I) has reversed shading (dark for low values) compared to the other subplots. It would be easier to interpret these graphs if their color coding was more consistent.

– Page 18: Similar to the difficulty of interpreting the TD error results, it is not clear what a "good" or "bad" KL divergence from the neural data would be. Any hypotheses on how to ground the numbers provided here would help to improve the quality of the results.

– Page 20: It is mentioned that the predictive timescale may be a separate gain term which the hippocampus takes as input, but there is evidence that different regions of the hippocampus seem to operate on different timescales. See for example: https://www.jneurosci.org/content/42/2/299.abstract. Is there a way to reconcile these ideas?

– Page 23: Section 4.5 describes the procedure for learning the parameters of the weight update rule as CMA-ES. Mentioning the fact that an evolutionary algorithm is used for learning these weights would help to make Section 2.3 more clear.

– Figures 5D-E and similar supplementary figures: if there is a parameter region that is unexplored then the color used for such region should be outside of the colormap. One of the reviewers suggests replacing white with gray for such region in these figures.

– Line 173: the text makes the distinction between an "SR-like" representation, and an "exact SR". What is the difference? Why is it important to have an exact of the SR in the neural activity, rather then eg a monotonic encoding of the SR?

– The RNN described in Equation 2 is not of the standard form (the non-linearity is applied after the connectivity matrix, ie f(J x) instead of Jf(x)). Is this detail important? If not, why not use the more standard form to avoid confusion?

– A line of work in the Fusi lab has examined plasticity rules that lead to the encoding of transition probabilities (eg Fusi et al., Neuron 2007). In particular, a paper by the reviewing editor (Ostojic and Fusi Front Comp Neuro 2013) examined the encoding of transition probabilities using plasticity rules that look similar to this manuscript. This is mentioned just for information, the authors should decide if those papers are relevant.

– Figures 5D-E and similar supplementary figures: if there is a parameter region that is unexplored then the color used for such region should be outside of the colormap. One of the reviewers suggests replacing white with gray for such region in these figures.

---

## [Author Response]

Main Comments:1) The form of the plasticity rule in Equation 4 is motivated by the requirement that synaptic weights encode a properly normalised transition probability matrix (lines 92-96). But why is the normalisation important? What would change if synaptic weights were simply monotonic functions of transition probabilities, without normalisation? Presumably that would allow for a broader range of plasticity rules.

The reviewer makes an interesting point that a range of possible rules may yield useful representations even if they do not learn the transition probability matrix exactly. We tested these ideas (see below) and found that normalization is generally important for maintaining stable dynamics in the recurrent network. Many forms of normalization can learn predictive representations similar to the SR, as long as the normalization is performed across rows of the weight matrix independently. We have added a few sentences of text (lines 107-116) and a supplementary figure summarizing and showing these results. The details of our additional analyses and the text added to the manuscript are given below.

We constructed representations using different normalization procedures. For these tests, we simulated a random walk on a circular track, as in Appendix 7-figure 1B, for 10 minutes of simulation time.

If the synaptic weight matrix estimates the normalized transition probability matrix (as in equation 4), the resulting SR matrix over the course of the walk is as Appendix 7-figure 1A.

As an initial test of the role of normalization, we did the same simulation, removing normalization. Thus, $J$ will be equal to the count of observed transitions, i.e. $J_{ij}$ is equal to the number of experienced transitions from state $j$ to state $i$. We refer to this as a count matrix. Note that, without normalization, the values in the count matrix increase steadily over the course of the simulation. This quickly results in unstable dynamics due to the weights of the matrix being too large. We can quantify this instability by plotting the maximum eigenvalue of the weight matrix, where an eigenvalue >= 1 indicates instability (Sompolinsky et al., 1988), Appendix 7-figure 1B.

A simple way to prevent instability is to scale the weights of the matrix by a constant factor such that the dynamics are in the stable regime (specifically, ensuring that the maximum eigenvalue of the synaptic weight matrix is below 1). A careful choice of a scaling value can ensure network activity remains stable within this walk, although this is not a generalizable solution as different scaling values will be needed for different random walks and tasks.

Informed by Appendix 7-figure 1C, we chose a scaling factor of 1.75. and tested what neural representations look like throughout the walk.

Compared to the ground truth SR, the representations above are not very predictive (the off-diagonal elements of the SR are not captured well), and the activity strength seems to be unevenly distributed across states. For instance, there is more activity at state 8 than other states. It is likely that depressing all synapses by the same factor (similar in flavor to Fiete et al., 2010) does not correct for differences in occupancies. In other words, states that happen to be visited more by chance are likely to dominate the dynamics, even if the transition statistics are identical across all states (Appendix 7-figure 1D).

As a further test of different ways of parameterizing the synaptic weight matrix, we can instead scale the count matrix in a row-by-row fashion. Specifically, we divide each row $i$ of the count matrix by the maximum of row $i$ (and some global scaling factor to ensure stability). Note that this is in contrast to $T$, where each row is divided by its sum. The resulting steady state activity matrices (Appendix 7-figure 1E).

This is closer to the SR matrix expected if the synaptic weight matrix estimates $T$. There is a slight unevenness early on in learning the diagonal of the matrix. However, given enough observed transitions, the predictive representation looks reasonable and quite similar to the SR matrix.

Overall, we see that the intuition of the reviewer is correct: there are likely other formulations of the synaptic weight matrix that can give a representation similar to the SR. The important ingredient for this to happen appears to be some form of row-dependent normalization—that is, neurons should have their synaptic weights normalized independently of each other. This ensures that occupancy is not conflated with transition statistics. We added the figures in appendix 7. We also added the following passage to the introduction of the plasticity rule in Results section 2.2 (and use the suggested reference to Fiete 2010):

“The second term in equation 4 is a form of synaptic depotentiation. Depotentiation has been hypothesized to be broadly useful for stabilizing patterns and sequence learning [37, 49], and similar inhibitory effects are known to be elements of hippocampal learning [50, 51]. In our model, the depotentiation term in equation 4 imposes local anti-Hebbian learning at each neuron—that is, each column of $J$ is normalized independently. This normalizes the observed transitions from each state by the number of visits to that state, such that transition statistics are correctly captured. We note, however, that other ways of column-normalizing the synaptic weight matrix may give similar representations (Figure S1).”

2) As the results of the paper strongly rely on the normalizing term in Equation 4. one of the reviewers suggests potentially moving upfront part of the discussion of this term, and enlarging the paragraph that discusses the biological plausibility of this specific term. Clearly laying out, for the non-expert reader, why it is biologically plausible compared to other learning rules. Also consider moving the required material to establish the novelty of such term: a targeted review of the relevant literature (current lines 358-366 and 413-433). This would allow the reader to understand immediately the significance and relative novelty of such term. For example, this reviewer personally wondered while reading the paper of how different was such term from the basic idea of Fiete et al., Neuron 2010 (DOI 10.1016/j.neuron.2010.02.003).

The reviewer points out that more context and clarity around the plasticity rule would be useful, particularly since an understanding of the plasticity rule is integral to the paper.

We would like to clarify that, although the RNN-S is more biologically plausible than the FF-TD learning rule, there is likely additional biological complexity/realism that can be added to the RNN-S learning rule. We wanted to find the simplest rule that could capture the essence of the SR, which is why we focused on the particular form of the learning rule we used.

Aside from plausibility, a key aspect of the RNN-S normalizing term (as discussed in Main Comment 1) is that it independently normalizes each column of the synaptic weight matrix. This is in contrast, say, to the Fiete et al., 2010 paper (which has a global depressive term) and other similar plasticity rules. We directly tested the effect of different normalizations on the RNN representations (see response to Main Comment 1), and find that column-specific normalization is important for capturing transition statistics.

To make these subtle points more clear to the reader, we added additional sentences about biological realism and other forms of learning rules in the section where the learning rule is introduced:

“Crucially, the update rule (equation 4) uses information local to each neuron (Figure 1h), an important aspect of biologically plausible learning rules.

The second term in equation 4 is a form of synaptic depotentiation. Depotentiation has been hypothesized to be broadly useful for stabilizing patterns and sequence learning [37, 49], and similar inhibitory effects are known to be elements of hippocampal learning [50, 51]. In our model, the depotentiation term in equation 4 imposes local anti-Hebbian learning at each neuron-- that is, each column of $J$ is normalized independently. This normalizes the observed transitions from each state by the number of visits to that state, such that transition statistics are correctly captured. We note, however, that other ways of column-normalizing the synaptic weight matrix may give similar representations (Figure S1).”

3) Related to the first point, the text insists on the fact that \γ is not encoded in the synaptic weights (eg line 89). Again, it is not entirely clear why this is important and justified, since γ is an ad-hoc factor in Equation 2. Presumably the proper normalisation of γ relies on the normalisation of J discussed above? It seems that this constraint could be relaxed.

The reviewer is correct that factorizing \γ as a separate factor from the synaptic weights (J) is a notational choice. We include the \γ as a factor distinct from the synaptic strengths for several reasons. The first is for consistency with previous literature. In the SR literature, \γ is factorized out of normalized transition matrices. Similarly, in RNN literature, it is typical to analyze a global gain factor (g), which determines the operating regime of the network (Sompolinsky et al., 1988). Our second is mechanistic. We interpret \γ as a measure of the gain of the network units, that is, as a physiological property of the neurons. The synaptic matrix J, on the other hand, measures the strengths of synapses. Keeping them separate allows for more flexibility, which leads to the third reason. Treating \γ as a separate factor allows the network to retrieve successor representations of different predictive strengths. Importantly, this dynamic predictive ability is achieved without changing any synaptic weights and without additional learning. In other words, a separate \γ allows us to decouple the learning and retrieval processes, providing more flexibility in using the SR.

To make this point more clearly in the text, we added the following lines to explain this rationale:

“Here, the factor $\γ$ represents the gain of the neurons in the network, which is factored out of the synaptic strengths characterized by J. Thus, $\γ$ is an independently adjustable factor that can flexibly control the strength of the recurrent dynamics (see [46]). A benefit of this flexibility is that the system can retrieve successor representations of varying predictive strengths by modulating the gain factor $\γ$. In this way, the predictive horizon can be dynamically controlled without any additional learning required.”

4) As a consequence of the body of the text being devoted to the analysis of the design choices behind the proposed model, a relatively smaller portion of the work involves direct comparisons with neural data. In these comparisons, while it is apparent that there is a reasonable match between the proposed model and the empirical data, it is difficult to interpret these results. This is because it is unclear what should be expected of a good or bad model given the data being analyzed (TD error and KL divergence), and reasonable baselines to compare against are not presented outside of the traditional TD algorithm, which is shown to be comparable to the proposed RNN based method in a number of cases.

The reviewer suggests conducting control analyses to help with interpretability of the TD error and KL divergence results. Specifically, they suggest comparing the performance of the RNN and FF network to good and bad models.

As an example of a “bad” model, we calculate the TD error and KL divergence of a feedforward network with weights randomly drawn from the distribution of weights of the FF-TD model at the end of learning. We call this the Shuffle model, and it is representative of a model without learned structure but with a similar magnitude of weights as the FF-TD model. Specifically, the shuffle model has much higher TD error than the RNN or FF network.

As an example of a “good” model, we calculate the KL divergence between randomly split halves of the dataset from Payne et al., (2021). Specifically, we compare the place field statistics of a random half of the neurons from Payne et al., (2021) with another random half. We calculate the KL divergence between the distributions calculated from each random half. This is repeated 500 times. We call this “Data” in the plot, and it is representative of a lower bound on KL divergence. Intuitively, it should not be possible to fit the data of Payne et al., as well as the dataset itself can. We compare this KL divergence to the KL divergence between each model and the neural data.

The KL divergence of the shuffle model is quite close to the FF network, suggesting that most of the “place-like” qualities of the FF network are more likely a reflection of the input features being “place-like” than the learned weights constructing these place fields. Place fields from the RNN-S model are more similar to neural data than the FF-TD or the Shuffle models. The analysis on split halves of the Payne et al., dataset shows a low KL divergence (specifically, around 0.12) that is much smaller than any of the models.

We add the two plots into the supplementary material (Figure S). Additionally, we report these results in the following sections of the main text:

In the section introducing TD error:

“We want to compare the TD loss of RNN-S to that of a non-biological model designed to minimize TD loss… Both models show a similar increase in TD loss as $\γ_R$ increases, although the RNN-S has a slightly lower TD loss at high $\γ$ than the FF-TD model. Both models perform substantially better than a random network with weights of comparable magnitude (Figure S5d).”

In the section introduced summed KL divergence:

“We combined the KL divergence of both these distributions to find the parameter range in which the RNN-S best fits neural data (Figure 6g). This optimal parameter range occurs when inputs have a spatial correlation of $\σ \approx 8.75$ cm and sparsity $\approx 0.15$. We note that the split-half noise floor of the dataset of Payne et al., is a KL divergence of $0.12$ bits (Figure S6E).

…

We next tested whether the neural data was better fit by representations generated by RNN-S or the FF-TD model. Across all parameters of the input features, despite having similar TD loss (Figure 5de), the FF-TD model has much higher divergence from neural data (Figure 6gi, Figure S6), similar to a random feedforward network (Figure S6E).”

5) It would be useful to have a “limitations” paragraph in the discussion clearly outlining what this learning rule couldn’t achieve. For example, Stachenfeld et al., Nat.Neuro. have many examples where the SR is deployed. Does the learning rule suggested by the authors would always work across the board, or are there limitations that could be highlighted where the framework suggested would not work well. No need to perform more experiments/simulations but simply to share insight regarding the results and the capability of the proposed learning rule.

The reviewer suggests adding further conceptual clarity to the discussion by giving insight into the limitations of the model and its differences from the Stachenfeld et al., simulations. We anticipate that the RNN-S can capture the results seen in Stachenfeld et al., since the SR learned by the algorithm in Stachenfeld et al., is identical to the representation learned by the RNN-S in the one-hot case.

However, using a recurrent architecture does impose limitations on how the learning rule is structured and how the network can be used. In particular, care must be given to avoid instability in the network due to build-up of recurrent activity. We proposed a ‘learning’ and ‘retrieval’ mode in the network precisely to control such instabilities. Furthermore, the recurrency of the network means that errors in transition structure can compound across a long horizon of prediction. This is especially problematic in the case of non-one-hot features, where greater errors in transition estimation are likely for more densely coded features.

We added a limitations paragraph in the discussion focusing on the limitations of using a recurrent network model (as opposed to a feedforward network). We also further tie these observations to biological evidence:

“There are inherent limitations to the approach of using a recurrent network to estimate the SR. For instance, network dynamics can be prone to issues of instability due to the recurrent buildup of activity. To prevent this instability, we introduce two different modes of operation, ``learning’’ and ``retrieval’’. An additional limitation is that errors in the estimated one-step transition can propagate over the course of the predictive rollout. This is especially problematic if features are more densely coded or more correlated, which makes one-step transition estimations more difficult. These insights into vulnerabilities of a recurrent network have interesting parallels in biology. Some hippocampal subfields are known to be highly recurrent [92, 93, 94, 95]. This recurrency has been linked to the propensity of the hippocampus to enter unstable regimes, such as those that produce seizures [96, 97, 98, 99]. It remains an open question how a healthy hippocampus maintains stable activity, and to what extent the findings in models such as ours can suggest biological avenues to tame instability.”

Other comments/suggestions:– Page 1: The introduction motivates this work with a discussion of hippocampal memory (storage and retrieval), but the work focuses on the SR which is inherently prospective. The first paragraph of the text could be revised to better make this connection beyond simply stating that the hippocampus is involved in both memory and future prediction.

The reviewer makes an Important point, and in fact the connection between episodic memory and predictive maps in the hippocampus is an active area of research in the hippocampus field. We have edited the first paragraph of the introduction to better explain and flesh out the hypothesized connections between predictive coding in the hippocampus and its function in memory:

*“*To learn from the past, plan for the future, and form an understanding of our world, we require memories of personal experiences. These memories depend on the hippocampus for formation and recall [1, 2, 3], but an algorithmic and mechanistic understanding of memory formation and retrieval in this region remains elusive. From a computational perspective, a key function of memory is to use past experiences to inform predictions of possible futures [4, 5, 6, 7]. This suggests that hippocampal memory is stored in a way that is particularly suitable for forming predictions.”

– Page 2: The end of the introduction would be stronger if the motivation for an RNNs usage was tied to the literature on the known recurrent dynamics of the hippocampus. See for example: https://www.frontiersin.org/articles/10.3389/fncel.2013.00262/full

https://www.frontiersin.org/articles/10.3389/fncel.2013.00262/full

We add the following sentences in the introduction to further motivate the usage of RNNs with known hippocampal anatomy/dynamics, adding in the suggested reference:

“A promising direction towards such a neural model of the SR is to use the dynamics of a recurrent neural network (RNN) to perform SR computations [39, 40]. An RNN model is particularly attractive as the hippocampus is highly recurrent, and its connectivity patterns are thought to support associative learning and recall [41, 42, 43, 44]. However, an RNN model of the SR has not been tied to neural learning rules that support its operation and allow for testing of specific hypotheses.”

– Page 6: It is not clear the extent to which the FF-TD model differs from a canonical tabular SR algorithm or linear SF algorithm. My understanding is that it is the same, but the presentation in Figure 1i for example makes this somewhat unclear.

Yes, you’re correct that the FF-TD model is exactly the linear SR/SF algorithm. We’ve added clarifying sentences in the section introducing the FF-TD model to emphasize this:

*“*As an alternative to the RNN-S model, we consider the conditions necessary for a feedforward neural network to compute the SR. Under this architecture, the $M$ matrix must be encoded in the weights from the input neurons to the hidden layer neurons (Figure 1g). This can be achieved by updating the synaptic weights with a temporal difference (TD) learning rule, the standard update used to learn the SR in the usual algorithm… The FF-TD implements the canonical SR algorithm.”

– Pages 6 – 11: it may be of benefit to more strongly support the various modifications to the model with connections to known or hypothesized hippocampal neural dynamics.

The reviewer suggests motivating the modifications to the model by connecting to known biological mechanisms. We currently make these connections in the Discussion section. To make these points earlier in the paper, we summarize some of these points made in the discussion and put them earlier in the paper. Specifically, we make the following additions:

In the section introducing the anti-Hebbian term:

“The second term in equation 4 is a form of synaptic depotentiation. Depotentiation has been hypothesized to be broadly useful for stabilizing patterns and sequence learning [37, 49], and similar inhibitory effects are known to be elements of hippocampal learning [50, 51].”

In the section introducing the adaptive learning rate:

“If the learning rate of the outgoing synapses from each neuron $j$ is inversely proportional to nj(η=1nj(t)), the update equation quickly normalizes the synapses to maintain a valid transition probability matrix (Supplementary Notes 4). Modulating synaptic learning rates as a function of neural activity is consistent with experimental observations of metaplasticity [56, 57, 58]”

In the section adding a nonlinearity into network dynamics:

“One way to tame this instability is to add a saturating nonlinearity into the dynamics of the network. This is a feature of biological neurons that is often incorporated in models to prevent unbounded activity [60]. Specifically, instead of assuming the network dynamics are linear ($f$ is the identity function in equation 2), we add a hyperbolic tangent into the dynamics equation.”

– Page 14: It states that“"We discretize the arena into a set of states and encode each state as a randomly drawn feature ϕ”" If I understand correctly, these features are not completely random, and instead follow the distribution described in Section 2.5. As it currently reads, it seems that these features might be drawn from a uniform random distribution, which would be misleading.

Thank you for pointing this ou— it is true that the features in Section 2.6 are constructed the same as in Section 2.5. We’ve made this more explicitly clear in Section 2.6:

*“*We discretize the arena into a set of states and encode each state as in Section $2.5$.”

– Page 14: In Section 2.6 there is an assumption that a certain level of TD error corresponds to good performance. It is not clear what should objectively be considered a reasonable TD error. This is especially difficult to interpret in the case where both the RNN-S and FF-TD models display comparable performance. Is there perhaps some other baseline you would expect to perform considerably worse?

This comment is already raised and addressed in Main Comment 4.

– Page 17: In Figure 4 it is somewhat confusing that the KL divergence (subplots G and I) has reversed shading (dark for low values) compared to the other subplots. It would be easier to interpret these graphs if their color coding was more consistent.

Thanks for the clarifying suggestion. We reversed the color map for KL divergence.

– Page 18: Similar to the difficulty of interpreting the TD error results, it is not clear what a“"goo”" or“"ba”" KL divergence from the neural data would be. Any hypotheses on how to ground the numbers provided here would help to improve the quality of the results.

This comment is already raised and addressed in Main Comment 4.

– Page 20: It is mentioned that the predictive timescale may be a separate gain term which the hippocampus takes as input, but there is evidence that different regions of the hippocampus seem to operate on different timescales. See for example: https://www.jneurosci.org/content/42/2/299.abstract. Is there a way to reconcile these ideas?

The reviewer points out an interesting hippocampal finding that is not obviously explained in the RNN-S model: some anatomical axes of the hippocampal formation appear to contain a continuum of predictive timescales in their neural activity (Dolorfo 1998, Brun 2008). This is a well-supported finding with interesting functional implications, and thus is worth discussing and addressing in the paper.

One way to model an anatomical gradient of predictive timescales is to use a series of RNN-S systems. Each of these systems would have a different $\γ$ values that is used during retrieval. Thus, despite these systems receiving the same feature inputs, each network can estimate the state of the animal across different timescales.

Alternatively, the gradient in timescales or granularity could exist on the input level. As in the first hypothesis, we can assume a series of RNN-S systems, except all systems utilize the same $\γ$ value during retrieval. If each system receives inputs that encode different granularities of the animal’s state space (in a spatial example: perhaps one set of inputs uses a state space that divides the arena into quadrants, while another set of inputs uses a state space that divides the arena into a 10x10 grid), then each RNN-S network will naturally develop representations across a continuum of scales.

Both these hypotheses can be functionally useful as a way to learn hierarchical structure and use that information for planning.

We choose to emphasize the first hypothesis (a gradient of $\γ$ values), and summarize this idea by add the following sentences into the discussion paragraph on flexible $\γ$:

*“*The idea that the hippocampus might compute the SR with flexible $\γ$ could help reconcile recent results that hippocampal activity does not always match high-$\γ$ SR [79, 80]. Additionally, flexibility in predictive horizons could explain the different timescales of prediction observed across the anatomical axes of the hippocampus and entorhinal cortex [88, 89, 90, 91, 92]. Specifically, a series of successor networks with different values of γ used in retrieval could establish a gradient of predictive timescales. Functionally, this may allow for learning hierarchies of state structure and could be useful for hierarchical planning [93, 94, 95].”

– Page 23: Section 4.5 describes the procedure for learning the parameters of the weight update rule as CMA-ES. Mentioning the fact that an evolutionary algorithm is used for learning these weights would help to make Section 2.3 more clear.

We added additional sentences in Section 2.3 clarifying how parameters were learned:

*“*To systematically explore the space of plasticity kernels that can be used to learn the SR, we performed a grid search over the sign and the time constants of the pre -> post and post -> pre sides of the plasticity kernels. For each fixed sign and time constant, we used an evolutionary algorithm to learn the remaining parameters that determine the plasticity kernel.”

– figures 5D-E and similar supplementary figures: if there is a parameter region that is unexplored then the color used for such region should be outside of the colormap. One of the reviewers suggests replacing white with gray for such region in these figures.

Thanks for the clarifying suggestion. We switched the color of the unexplored region from white to gray.

– Line 173: the text makes the distinction between an“"SR-lik”" representation, and an“"exact S”". What is the difference? Why is it important to have an exact of the SR in the neural activity, rather then eg a monotonic encoding of the SR?

By “exact SR”, we mean the error between the steady state dynamics matrix and the SR matrix is precisely zero. “SR-like” was our loose way of referring to representation matrices with some amount of mean absolute error from the SR matrix that was still seemingly minimal but not zero.

The reviewer raises an important question about how crucial it is for the network to learn the SR exactly, versus other representations that may also capture long-horizon predictions.

This is similar in spirit to the question raised in Main Comment 1. We showed in an analysis for Main Comment 1 that it may not be necessary to learn the SR exactly, and that a range of possible rules may yield similar representations. The SR is convenient as a reasonable formalization of long-horizon predictions. The analysis referenced in line 173 (Figure 2J) also shows that plasticity kernels with varying time constants yield representations that are similar to the SR.

We clarify the statement previously in line 173 to remove the term “is SR-like” with “has minimal error from the SR matrix”. We also emphasize that the results of the analysis further supports that many predictive representations look similar to each other, and that an exactathemaatical equivalence to the SR is not the most important aspect of a predictive representation:

*“*Finally, we see that even plasticity kernels with slightly different time constants may give results with minimal error from the SR matrix, even if they do not estimate the SR exactly (Figure 2j). This suggests that, although other plasticity rules could be used to model long-horizon predictions, the SR is a reasonable—-though not strictly uniqu— model to describe this class of predictive representations.”

– The RNN described in Equation 2 is not of the standard form (the non-linearity is applied after the connectivity matrix, ie f(J x) instead of Jf(x)). Is this detail important? If not, why not use the more standard form to avoid confusion?

The reviewer points out that our equation uses Δx=−x+f(Jx)+i. It is standard to use either a firing rate representation (Δr=−r+f(Jr+i)) or a voltage representation (Δv=−v+Jf(v)+i). Choosing between the firing rate or voltage representation is not critical. Indeed, if input is not considered, these forms are equivalent up to a transformation (Miller and Fumarola 2012).

Nevertheless, the reviewer is correct that we used a non-conventional amalgamation of these two standard forms.

We re-ran analyses using Jf(x) instead of f(J x). We find that there is no obvious difference in the results generated. We updated the equation in the text to match the standard form. We then updated Figures 3, 5, 6, S3, S5, S6 to reflect this change in the model.

– A line of work in the Fusi lab has examined plasticity rules that lead to the encoding of transition probabilities (eg Fusi et al., Neuron 2007). In particular, a paper by the reviewing editor (Ostojic and Fusi Front Comp Neuro 2013) examined the encoding of transition probabilities using plasticity rules that look similar to this manuscript. This is mentioned just for information, the authors should decide if those papers are relevant.

Thanks for the recommendation— the Ostojic and Fusi paper is indeed quite relevant. It seems like equation 2 of the Ostojic and Fusi paper is the same as our plasticity rule under one-hot encoding assumptions. In the case with more complex input features, the Ostojic and Fusi paper would be identical to the “Independent Normalization” model we present as comparison in Figure 4.

Overall, it is promising and exciting to find another study that arrives at similar conclusions: “Our study shows that synapses encode transition probabilities under general assumptions and this indicates that temporal contiguity is likely to be encoded and harnessed in almost every neural circuit in the brain.” (from the abstract of Ostojic and Fusi).

We have updated the discussion to include this reference:

“Estimating $T$ directly provides RNN-S with a means to sample likely future trajectories, or distributions of trajectories, which is computationally useful for many memory-guided cognitive tasks beyond reinforcement learning, including reasoning and inference (Ostojic et al., 2013, Goodman et al., 2016). The representation afforded by $T$ may also be particularly accessible by neural circuits. Ostojic et al., (2013) note that only few general assumptions are needed for synaptic plasticity rules to estimate transition statistics. Thus, it is reasonable to assume that some form of transition statistics are encoded broadly across the brain.”

– Figures 5D-E and similar supplementary figures: if there is a parameter region that is unexplored then the color used for such region should be outside of the colormap. One of the reviewers suggests replacing white with gray for such region in these figures.

This is a duplicate comment of another comment and has been fixed.